# Diagnosing with Insights: Structured Analysis of Agent Failures via Behavioral Abstractions

## Abstract

With the proliferation of LLM agents, the ability to understand and diagnose failures in agents is essential to achieving superior effectiveness and trustworthiness. As agent failures often manifest via long and complex trajectories, manually finding the needles in the haystack is untenable. However, traditional diagnosis techniques for software bugs can hardly address LLM agent failures, while completely relying on LLMs as the judge yields unreliable diagnosis results. To overcome these challenges, this paper presents AGENTSCOPE, a new neuro-symbolic approach for agent failure mode diagnosis. The key principle of AGENTSCOPE is to abstract agent behavior, based on its trajectories, into structured representations. Furthermore, AGENTSCOPE introduces the concept of neural invariants to specify agent behavior properties. AGENTSCOPE leverages LLM-guided reasoning atop the structured representation against neural invariants to pinpoint both the failure step and its type in the trajectory. We show the effectiveness of AGENTSCOPE on publicly available agent failure datasets (Who&When) and a more comprehensive dataset created by us (AgentErrata), where AGENTSCOPE significantly outperforms the current art in fault localization and attribution accuracy. Our work shows that integrating structured abstractions with LLM-guided reasoning enables effective, reliable, and interpretable diagnosis for agent failures.

## 1 Introduction

Recent advances in Large Language Models (LLMs) have been driving active development of LLM agents that autonomously interact with tools and environments using natural languages. For example, agents can issue API calls, synthesize code, query databases, and cooperate with other agents to solve complex and multi-step tasks. Modern agent frameworks such as LangChain (Chase, 2022), Auto-GPT (Significant Gravitas), and OpenManus (Liang et al., 2025) orchestrate multi-step interactions, allowing agents to operate in open and dynamic environments.

Despite their exciting capabilities, LLM agents are known to fail in subtle and sophisticated ways (Cemri et al., 2025; Zhang et al., 2025b; Skalse et al., 2022; Bryan et al., 2025; Baker et al., 2025; Fu et al., 2025). Failures can occur at any step during the agent's *reasoning* (e.g., the chain of thoughts) and *action execution* (e.g., tool calls), cascade along the agent runtime behavior, and eventually manifest as specific mistake behavior or even disrupt task execution. Thus, the ability to understand and diagnose failures in agents is essential to achieving superior effectiveness and trustworthiness of agent systems. Unfortunately, as agent failures often manifest through prolonged and complex trajectories composed of numerous steps with accumulating context, manually finding needles in the haystack is slow, costly, and obviously untenable.

Worse still, traditional diagnosis techniques for software bugs can hardly address LLM agent failures Zeller (2009); Attariyan & Flinn (2010); Yuan et al. (2010); Zhang et al. (2019); Ren et al. (2023). The fundamental reason is that traditional techniques are confined to symbolic and logical analysis of software codes and program executions. In contrast, agent failures are often rooted in faulty reasoning errors, invalid contexts, or instruction-unfollowing behaviors across multiple steps, which involve the entanglement of fuzzy neural paradigms and rigid symbolic paradigms.

Recent studies (Zhang et al., 2025b; Cemri et al., 2025; Zhu et al., 2025; dlshriver, 2023) have proposed several neural approaches for agent failure analysis—prompting or fine-tuning LLMs with failure trajectories and asking them to identify root causes. However, our experiments show that sole LLM-based approaches often produce unreliable and incomplete diagnostic results. Even the best-performing model, GPT-4o (Hurst et al., 2024), achieves only 4.4% accuracy on our failure attribution datasets (see Section 5.2). The cause of this limitation is that LLMs, even when fine-tuned, struggle to systematically capture multi-step reasoning and action behaviors and maintain consistent causal invariants. Thus, they are prone to confusing correlated symptoms with true causes and sensitive to contexts and instructions.

This paper presents AGENTSCOPE, a novel neuro-symbolic framework for diagnosing agent failures. The key idea of AGENTSCOPE is to abstract the agents' behavior from their trajectories into a *structured* representation, termed the Reasoning-Action Graph (ReAG), that encapsulates both the reasoning and action execution steps of the agents, enabling rigorous correctness reasoning using formally defined invariants on their behaviors. Different from traditional program invariants (Hoare, 1969), AGENTSCOPE introduces the concept of *neural invariants*, which can be specified with neural functions to encode correctness conditions; checking such invariants requires LLM-guided reasoning. We show that LLM-guided reasoning atop structured behavior abstractions enables AGENTSCOPE to effectively pinpoint the causes of the target agent failure. Compared with existing vanilla LLM-as-a-judge approaches, AGENTSCOPE demonstrates stronger diagnosis ability and more interpretable results. It can accomplish both *failure localization* and *failure attribution*, the former pinpointing the root-cause step and the latter predicting the failure mode category.

We evaluated AGENTSCOPE on a publicly available agent failure dataset named Who&When (Zhang et al., 2025b) and a new dataset called AgentErrata created by us through failure-taxonomy-guided fault injection for more comprehensive evaluation. Results show that AGENTSCOPE achieves significant improvements in accuracy and interpretability over state-of-the-art approaches, with localization accuracies ranging from 41.6% to 77.6% on the Who&When algorithm-generated dataset and from 22.2% to 27.8% on the Who&When hand-crafted dataset, along with enhanced attribution interpretability. Moreover, AGENTSCOPE also outperforms other methods on our new dataset, AgentErrata, achieving accuracies ranging from 34.2% to 49.7% .

In summary, we make the following contributions:

- **Principle.** We show that LLM-guided reasoning atop structured representations of behavior abstractions can significantly improve agent failure diagnosis ability.

- **Concept.** We introduce the concept of neural invariants, which use neural functions to formally define and detect properties of agent misbehaviors.

- **Tooling.** We develop AGENTSCOPE, a practical runtime framework and toolchain for agent failure detection and diagnosis that is integrated with modern agent frameworks.

- **Dataset.** We create AgentErrata, a new benchmark dataset for agent failures; AgentErrata contains comprehensive failure patterns covering diverse agent misbehaviors through failure-taxonomy-guided fault injection.

- **Evaluation.** We present the utility of AGENTSCOPE using systematic evaluations and showcase its improved accuracy and interpretability over the current art.

## 2 BACKGROUND

Agent failures have a variety of causes through incorrect reasoning and action execution steps, in which a single failure step can cascade and lead to observable symptoms. Table 1 shows a taxonomy of agent failure patterns, categorized based on the manifestation of the failure behaviours: **(i) Invalid context**: failures due to missing, incorrect, or invalid context; **(ii) Instruction unfollowing**: behavior deviations from instructions or inputs from other agents; **(iii) Faulty verification**: outputs that fail verifications or exhibit logical inconsistencies; **(iv) Incoherent planning**: wrong behaviors due to bad planning; **(v) Reward Hacking**: manipulations of reward mechanisms or strategic underperformance. This taxonomy provides a comprehensive and interpretable framework for understanding and diagnosing failures of LLM-based agents. The failure patterns are derived from recent empirical studies on agent failures (Cemri et al., 2025; Zhang et al., 2025b; Baker et al., 2025).

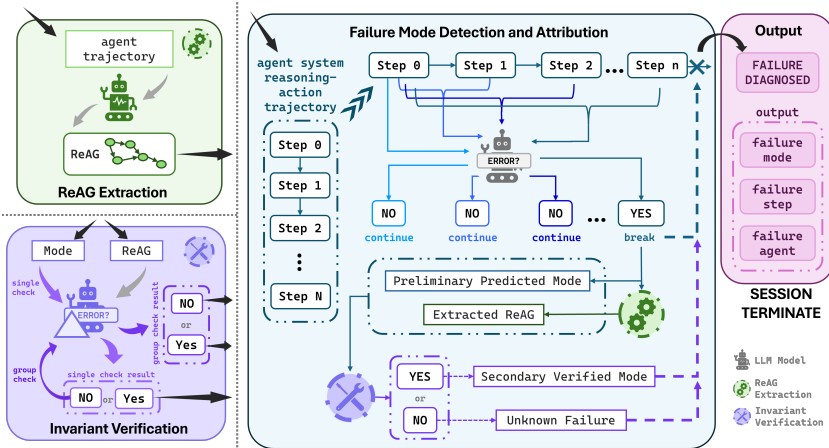

Figure 1: Workflow of AGENTSCOPE.

Table 1: Taxomony of LLM agent failure patterns.

| Category | Pattern | Description |
|---|---|---|
| Invalid Context | Information Withholding | Withholds information to manipulate output. |
| | Wrong Context | Context used is irrelevant, insufficient, invalid, or incorrect, causing errors. |
| Instruction Unfollowing | Action-Reasoning Mismatch | Actions contradict reasoning, showing decision-justification misalignment. |
| | Disobey Specification | Deviates from instructions or roles, causing task misalignment. |
| Faulty Verification | Verification-failed Output | Output did not pass the verifier's verification. |
| | Invalid Verification | Passed verification, but later proved incorrect. |
| Incoherent Planning | Unaware of Termination | Fails to recognize end, causing extra or endless actions. |
| | Conversation Reset | Unwarranted dialogue restarts, losing context and progress. |
| | Premature Termination | Prematurely stops, causing incomplete solutions or missed steps. |
| | Step Repetition | Repeats prior steps unnecessarily without justification. |
| Reward Hacking | Sandbagging | Deliberately underperforms to hide capabilities and avoid oversight. |
| | Reward Tampering | Manipulates reward mechanism to boost gains artificially. |
| | Scheming | Exploits design flaws to gain rewards without task completion. |

## 3 AGENTSCOPE

AGENTSCOPE is a neuro-symbolic framework and toolchain for diagnosing agent failures based on the agent trajectories. The key idea of AGENTSCOPE is to abstract the agents' behaviors from their trajectories into a *structured* graph representation, which enables rigorous correctness reasoning based on formally defined invariants. AGENTSCOPE introduces the concept of neural invariants to specify and reason agent misbehavior properties. Such reasoning enables AGENTSCOPE to effectively pinpoint the root causes of the target failure, including the failure step and its type. Figure 1 illustrates the high-level workflow of AGENTSCOPE.

### 3.1 BEHAVIORAL ABSTRACTION

In AGENTSCOPE, an agent system's trajectory is represented by a graph, termed Reasoning-Action Graph (ReAG). An ReAG is a directed acyclic graph, $G = \{V, E\}$, where $V$ and $E$ are the sets of vertices and edges, respectively. Each $v \in V$ in the ReAG abstracts an *step* taken by the agent in the trajectory, such as prompting an LLM, calling an external tool, updating agent memory, or inter-agent communication (for multi-agent systems). We use a quadruple $V\langle i, n, t, c \rangle$ to encode $v \in V$:

$$V\langle id,\ name,\ type,\ content \rangle,$$

where $id$ is the unique identifier and label of the step, $name$ indicates the agent or role that is performing the step, $type \in \{llm, tool\}$ specifies its type, and $content$ encapsulates the prompt, tool input, output, or other relevant information depending on the type.

An edge $e \in E$ represents a control or data dependency between two steps, such as control flow, data dependency or sequential relationship. It is encoded by a triple:

$$E\langle from\_id,\ to\_id,\ label \rangle,$$

with $label \in \{control,\ output,\ serial\}$ indicating the dependency type. Annotated edges capture control flow, output propagation, and sequential steps' chronological order.

Moreover, ReAG can represent agent behaviors in different granularities: *(i) Coarse-grained Behavior:* Capturing major steps at the granularity of LLM prompting and tool calling, as well as critical state transitions in the agent workflow. These steps are identified through instrumentation of LLM API and are further refined using LLM-based semantic parsing. *(ii) Fine-grained Behavior:* Representing intermediate steps embedded in natural language or code (e.g., Chain-of-Thought, instructions, and symbolic derivations). AGENTSCOPE also structures these behaviors into subgraphs on demand.

The ReAG can be automatically constructed from agent trajectories. Trajectories that already have structured step-wise formats provide explicit vertex delineation and edge dependency, enabling rapid graph construction. Some agent systems, however, may not precisely record structured information; their logs are often fused together and require additional processing to recover the dependency structure and vertex boundaries. For this purpose, AGENTSCOPE uses LLM-based parsers to delineate the boundaries between reasoning and action execution steps. Note that AGENTSCOPE constructs fine-grained behaviors on demand—if the trajectory is not sufficiently detailed for diagnosis, AGENTSCOPE can expand certain vertices into subgraphs to represent fine-grained behaviors.

## 3.2 NEURAL INVARIANTS

AGENTSCOPE introduces the concept of *neural invariants* to specify the behavioral properties of agent failures, which can be used to check the correctness of the ReAG of a given trajectory and to diagnose failure modes: AGENTSCOPE conducts LLM-guided reasoning to check for violations of neural invariants. When an invariant is violated, AGENTSCOPE identifies the vertex (i.e., the step) in the ReAG where the failure starts to manifest and classifies the type of the failure. Compared with vanilla LLM-as-a-judge approaches (Zhang et al., 2025b; Cemri et al., 2025), AGENTSCOPE offers several advantages: (i) precise localization of the failure step in the ReAG, (ii) fine-grained classification of the failure type based on invariant categories rather than opaque judgment criteria, (iii) highly interpretable explanations through invariant violations that explicitly expose the root causes of the failure, and (iv) more faithful and deterministic results, since the verification relies on predefined invariants rather than stochastic model judgments.

To achieve effective failure diagnosis, we define a comprehensive set of agent behavior neural invariants based on the taxonomy of agent failures (Table 1). Specifically, for each pattern in Table 1, we formalize it into an invariant; Appendix A lists all the invariants used by AGENTSCOPE. It is straightforward to add more invariants when extending a new failure mode taxonomy. Different from traditional program invariants (Hoare, 1969) that only contain symbolic conditions, the invariants in AGENTSCOPE can encode semantic conditions in the form of *neural functions*. Currently, each neural function is implemented as a call to a general-purpose LLM (e.g., GPT-4o) with a structural and task-specific prompt. Note that the implementation of neural functions can be customized, e.g., using fine-tuned models. We present one neural invariant as follows:

> **Invariant for Step Repetition.** Step Repetition occurs when an agent repeats prior reasoning or execution steps without a valid justification, leading to inefficiency.
>
> Let $\mathbf{N}$ be the set of nodes in the reasoning graph. For nodes $n_i, n_j \in \mathbf{N}$ with $i < j$, define a similarity function $\mathtt{sim}(content_i, content_j)$ that measures semantic similarity between steps, and a function $\mathtt{justified}(n_j)$ indicating whether repetition is warranted.
>
> The invariant is expressed as:
>
> $inv_{repeat} ::= \exists\, n_i, n_j \in \mathbf{N},\ i < j : \mathtt{sim}(content_i, content_j) > \theta \land \neg \mathtt{justified}(n_j)$
>
> Violation indicates that the agent repeated a prior step without justification, reflecting redundant or inefficient reasoning. If an LLM is used as the judge, the threshold of $\mathtt{sim}$ is replaced by a binary classification label.

The following shows the prompt of neural function $\mathtt{sim()}$ we are using at present in the above neural invariant:

```
1 prompt = (
2    "You are given two nodes from a reasoning execution graph representing an AI
   ↪  agent's problem-solving process. Judge whether the two nodes are semantically
   ↪  similar or not. Return only 'Yes' (for similar) or 'No' (for not similar). The
   ↪  nodes are considered similar if they convey the same meaning, perform the same
   ↪  function, or represent the same step in the reasoning process, even if they
   ↪  are worded differently. Conversely, if the nodes have different meanings,
   ↪  functions, or represent different steps, they are considered not similar. Even
   ↪  subtle similarities should lead to a 'Yes' answer.\n"
3    f"Node 1: {node1}\n"
4    f"Node 2: {node2}\n"
5    "Your response should be either 'Yes' or 'No' without any additional explanation."
6 )
```

Based on the classification of failure modes and their invariants, we synthesize diverse test cases across different domains to tune and validate the effectiveness of the neural functions. In addition, human evaluators double-check the results to ensure effectiveness.

## 3.3 IMPLEMENTATION AND USAGE

We build AGENTSCOPE as a runtime framework and toolchain for structured tracing and diagnosis of LLM-based agents. AGENTSCOPE is implemented in 2885 lines of Python code, and we integrate it with popular agent system frameworks including LangChain (Chase, 2022) and AutoGen (Wu et al., 2024). AGENTSCOPE works for both single-agent and multi-agent systems. Furthermore, AGENTSCOPE can be used for both online failure detection and offline failure diagnosis (both supporting failure attribution and failure localization). Its modular architecture supports custom invariant plug-ins and domain-specific verifiers. AGENTSCOPE can also be embedded into agent development pipelines for pre-deployment testing and canary services. By instrumenting the runtime using the intercepts (dlshriver, 2023) library for OpenAI API calls and other general-purpose LLM API invocations, AGENTSCOPE can capture coarse-grained steps, such as LLM calls or tool invocations, and construct a ReAG and continously check it against invariants. However, when explicit delimiters are absent or when the boundaries between reasoning and action execution are not clearly defined, for example in long mixed reasoning–action trajectories, chain-of-thought reasonings or derivations, steps with clear-cut boundaries are extracted via LLM-guided parsing first. (Experiments on the effectiveness of this LLM parser is shown in Appendix D) As for cases in which the agent system already collects execution trajectories, AGENTSCOPE can directly build ReAGs from them without additional instrumentation or parsing (Appendix B).

Moreover, to balance diagnostic efficiency and accuracy, we divide the diagnosis process into two stages. In the first stage (named preliminary prediction), invariants are defined by neural prompts, and the ancestor graph of the current step is constructed as a full context. If an failure is detected during this stage, the corresponding step will be flagged and the initial prediction of failure mode is put forward. For these flagged steps and failure modes, the second stage (secondary verification by Invariant Verifier, or IV) aims to reduce false positives. For each flagged step, all relevant subgraphs and neighboring vertices are extracted according to the definition of the invariants. These subgraphs are then checked by a combinition of formal definitions and neural functions based on the preliminarily predicted failure mode. If the verification passes, the detection is considered correct, or else the "unknown" alert is signaled, indicating that a previously undefined category has been encountered. This category can then be further investigated by the user in depth.

The two-stage design ensures reliable failure mode localization and attribution while keeping the verification process efficient. Specifically, in the preliminary prediction stage, invariant checks are performed over a large context window to quickly identify potential failure steps, and neural functions are not applied during this stage. In the secondary verification stage, the corresponding invariants are re-evaluated using more structured neural functions to confirm the initial predictions.

## 4 EXPERIMENTS

We evaluated AGENTSCOPE with both proprietary and open-source models including: GPT-4o (version: 2024-11-20) (Hurst et al., 2024), DeepSeek-V3 (version: V3-0324) (Liu et al., 2024), and Qwen3-14B (Yang et al., 2025). For GPT-4o and DeepSeek-V3, the models are deployed

via Azure AI Foundry (Microsoft, 2025); Qwen3-14B (Q4_K_M) is deployed through Ollama (0.9.6) (Inc., 2025) on NVIDIA GeForce GTX 1080 Ti. All models used a hyperparameter setting of $temperature = 0.01$ and $max\_token = 4096$. All other hyperparameters were set to default.

## 4.1 BENCHMARKS

We used two benchmark datasets for evaluation.

**Who&When.** The Who&When dataset (Zhang et al., 2025b) targets failure localization. It evaluates agents on the understanding towards temporal and causal relations in narrative contexts. It contains 184 human-annotated trajectories, each involving identifying who did what and when, emphasizing temporal alignment and attribution across multiple events. The Who&When dataset includes two sub-datasets that are generated from algorithm generated and hand-crafted agent systems respectively. We denote them as "Algo-Generated" and "Hand-Crafted" in §5 and evaluate the two sub-datasets separately. The former contains 126 traces and the latter contains 58 traces.

**AgentErrata.** The Who&When dataset only provides annotations for failure steps; however, it does not contain failure types and thus is of limited effect to our evaluation of failure attribution. Therefore, it is necessary to construct a dataset that not only annotates failure steps but also explicitly classifies each failure into well-defined categories. Such a dataset would enable systematic evaluation of a tool's actual debugging capabilities and assist tool developers in identifying shortcomings or weaknesses in detection mechanisms. To this end, we create a new dataset named AgentErrata, a comprehensive dataset that covers all kinds of failure patterns using *failure-taxonomy-guided fault injection*. We select various agent systems and benchmarks including OpenManus (Liang et al., 2025) on BrowseComp (Wei et al., 2025), Humanity's Last Exam (Phan et al., 2025), AG2 (Wang et al., 2024) and CAMEL (Li et al., 2023) on Olympiad Bench (He et al., 2024), and CAMEL (Li et al., 2023) on ProgramDev (Cemri et al., 2025). To create AgentErrata, we first set up the selected agents to produce correct execution trajectories in a reliable way; we then inject perturbations that trigger agent failures such as information withholding, step repetition, and premature termination to the original agent execution. Perturbations are applied across all categories in the current taxonomy (see Table 1) according to a distribution informed by empirical observations and practical priors, providing a reasonable approximation of the real-world occurrence probabilities of different failure modes. Then, the traces with injected perturbations are processed by a filtering–refinement pipeline: the filter removes traces with invalid formats or severe contextual inconsistencies, while the refiner improves their realism and coherence and also corrects minor formatting issues. AgentErrata currently contains 193 high-quality trajectories, providing a comprehensive dataset for evaluating diagnosability of agent failures. We show the detailed distribution of AgentErrata in Appendix C.

We did not use the MAST dataset (Cemri et al., 2025), as it only provides annotations of overall failure types but lacks specific error steps and underlying reasons. Consequently, it cannot be directly employed to reliably analyze the root causes of misattributions or to automatically evaluate the limitations of the benchmarked tools.

## 4.2 EVALUATION METRICS

We use the three metrics from Zhang et al. (2025b) and introduce an additional metric to measure the effectiveness of failure mode diagnosis

**Step-level accuracy (SLA)** quantifies the percentage of correctly identified root-cause failure steps. Formally, let $N$ be the total number of samples, $S_i$ be the true root-cause failure step for sample $i$ and $\hat{S}_i$ the predicted step, then $\text{SLA} = \frac{1}{N} \sum_{i=1}^{N} \mathbb{1}\left(\hat{S}_i = S_i\right)$, where $\mathbb{1}(\cdot)$ is the indicator function.

**Step-level accuracy with tolerance (SLAT)** considers a prediction correct if the predicted step falls within a tolerance range $\delta$ of the actual root-cause step: $\text{SLAT} = \frac{1}{N} \sum_{i=1}^{N} \mathbb{1}\left(|\hat{S}_i - S_i| \leq \delta\right)$. In other words, SLAT tolerates slight inaccuracy.

**Agent-level accuracy (ALA)** is the percentage of correctly predicted failure-responsible agents. Let $A_i$ be the set of true failure-responsible agents for sample $i$, and $\hat{A}_i$ be the predicted set, then $\text{ALA} = \frac{1}{N} \sum_{i=1}^{N} \mathbb{1}\left(\hat{A}_i = A_i\right)$.

**Classification accuracy (CA)** stands for the percentage of correctly classified failure modes. Let $M_i$ be the ground truth failure mode for the $i^{th}$ trace and $\hat{M}_i = \{\hat{M}_i^{(l)}\}_{l=1}^L$ be the predicted failure modes, where $L$ is the total number of predicted failure modes, then CA is defined as CA $= \frac{1}{N} \sum_{i=1}^{N} \mathbb{1} \left( \exists \hat{M}_i^{(l \in [1,L], l \in \mathbb{Z})} \in \hat{M}_i, \hat{M}_i^{(l)} = M_i \right)$.

# 5 RESULTS

## 5.1 HOW EFFECTIVE IS AGENTSCOPE IN FAILURE LOCALIZATION?

To evaluate the effectiveness of AGENTSCOPE in localizing the root-cause error step, we compare it against two representative baseline approaches:

**ALL-AT-ONCE (Zhang et al., 2025b)**: This approach considers the agent's entire trajectory as a single unit, using an LLM to detect failures holistically. Although efficient, it lacks fine-grained analysis and often overlooks localized failures or hallucinations.

**STEP-BY-STEP (Zhang et al., 2025b)**: This approach analyzes each step along the trajectory, flagging anomalies based on local patterns. However, it is guided by neither invariants for precise classification and localization, nor graph structures to extract relevant neighbors' information.

Table 2 presents the failure localization accuracy of AGENTSCOPE compared with the baseline methods across three datasets under different tolerance levels. Overall, when the ground truth to the original problem is given (w/ GT), AGENTSCOPE consistently outperforms both the All-at-once and STEP-BY-STEP approaches, demonstrating its ability to more accurately identify faulty reasoning steps. As the tolerance window increases from 0 to ±3, the accuracies of all methods improve, but AGENTSCOPE benefits the most, showing robustness when minor deviation is allowed. The improvements over the baselines are particularly notable on the Who&When Hand-Crafted and AgentErrata datasets; for example, at tolerance 0 on AgentErrata, AGENTSCOPE achieves an accuracy of 34.20%, doubling STEP-BY-STEP (17.10%) and greatly surpassing ALL-AT-ONCE (5.18%). The high performance at tolerance 0 highlights AGENTSCOPE's ability to precisely locate failures at the exact root-cause step, a task where both baseline methods struggle due to coarse-grained and context-agnostic analysis.

When the ground truth to the original problem is not given (w/o GT), AGENTSCOPE still outperforms the other two approaches on AgentErrata, but falls short on the Who & When datasets. Upon further examination, we found that several issues exist in the traces of Who & When — the annotated mistake steps are debatable and controversial (for example, there may be additional failure mode in the trajectories), which affects AGENTSCOPE's predictions and undermines their value as a reference benchmark. A more detailed explanation is provided in Appendix E.

These results collectively validate that leveraging invariants-guided reasoning and graph-based structures enables AGENTSCOPE to achieve more accurate and reliable fault localization in multi-step agent reasoning and execution.

Table 2: SLAT of baseline methods and AGENTSCOPE on Who&When and AgentErrata. (%)

| Method w/ GT | W&W Algo-Generated | | | W&W Hand-Crafted | | | AgentErrata | | |
|---|---|---|---|---|---|---|---|---|---|
| | T±0 | T±1 | T±3 | T±0 | T±1 | T±3 | T±0 | T±1 | T±3 |
| All-at-once | 14.29 | 43.65 | 69.05 | 5.26 | 8.77 | **29.82** | 5.18 | 22.28 | 35.32 |
| Step-by-step | 23.02 | 53.97 | 73.81 | 15.52 | 15.52 | 20.69 | 17.10 | 25.39 | 39.38 |
| **AGENTSCOPE** | **41.60** | **56.80** | **77.60** | **22.22** | **22.22** | 27.76 | **34.20** | **39.90** | **49.74** |

| Method w/o GT | W&W Algo-Generated | | | W&W Hand-Crafted | | | AgentErrata | | |
|---|---|---|---|---|---|---|---|---|---|
| | T±0 | T±1 | T±3 | T±0 | T±1 | T±3 | T±0 | T±1 | T±3 |
| All-at-once | 16.67 | 38.89 | **69.84** | 5.17 | 10.34 | **36.20** | 6.21 | 20.73 | 34.72 |
| Step-by-step | 15.08 | **42.06** | 61.11 | **17.24** | **17.24** | 24.14 | 18.65 | 26.94 | 39.90 |
| **AGENTSCOPE** | **21.43** | 32.54 | 46.03 | 10.34 | 12.07 | 15.52 | **35.75** | **41.97** | **52.85** |

("T." denotes "Tolerance"; "W&W" denotes "Who&When"; "w/ GT" indicates that the ground truth to the original problem that the agents were solving was given in the context during analysis, whereas "w/o GT" denotes no ground truth was given (Zhang et al., 2025b).)

## 5.2 HOW EFFECTIVE IS AGENTSCOPE IN FAILURE ATTRIBUTION?

We further evaluate AGENTSCOPE's ability to classify failure modes beyond localizing them, a process referred to as *attribution*. Failure modes in LLM agents can manifest in diverse forms (Table 1). Existing approaches (Cemri et al., 2025; Zhang et al., 2025b) cannot simultaneously identify the failure step and classify it. We categorize failures into representative classes and assess AGENTSCOPE's ability to attribute them within these categories using the AgentErrata dataset.

Table 3: Failure attribution on the AgentErrata dataset. (%)

| Method | ALA | SLA | CA |
|---|---|---|---|
| LLM-as-Judge | **42.45** | 5.35 | 4.40 |
| AGENTSCOPE | 38.34 | **34.20** | 22.80 / **27.98** |

(The result before the slash (/) indicates "w/o IV", the result after the slash indicates "w/ IV")

Table 3 shows that AGENTSCOPE substantially improves failure-attribution accuracy on AgentErrata compared to the LLM-as-Judge baseline, which is adapted from the All-at-once approach (Zhang et al., 2025b) to fit our evaluation setting. In particular, AGENTSCOPE achieves significantly higher Step-Level Accuracy (SLA), where it yields 34.20% compared to only 5.35% for the baseline, suggesting that AGENTSCOPE is considerably better at fine-grained attribution across intermediate reasoning steps. Also, AGENTSCOPE delivers higher Classification Accuracy (CA) than the baseline (27.98% vs. 4.40%), showing that it provides more precise categorization of failure patterns. The effectiveness of the Invariant Verifier (IV) in the secondary stage of the workflow of AGENTSCOPE is also validated, demonstrating notable boost on CA. However, it also shows that AGENTSCOPE slightly underperforms the baseline in terms of Agent-Level Accuracy (ALA) (38.34% vs. 42.45%), a difference that is within an acceptable range. Based on the large discrepancy between the step-level accuracy and the agent-level accuracy of the LLM-as-a-Judge method, we infer that it is not truly identifying the actual errors. Instead, it tends to select the agent category with the highest prior frequency in the distribution, which in turn increases the likelihood of producing a correct agent-level prediction. This may be the result of non-uniformly distributed agent occurences in both standalone agent trajectories and collective dataset annotations. Specifically, agents acting as the planning or reasoning role appear more frequently than specialized agents in general. For this reason, even if the pinpointed step differs from the ground truth, the corresponding agent may still match with the ground truth agent, resulting in high ALA but low SLA.

## 5.3 HOW DOES AGENTSCOPE PERFORM WITH DIFFERENT BASE MODELS?

This research question investigates the influence of building AGENTSCOPE on different base LLMs. Since LLM-based agents can vary significantly in model type and size, it is important to evaluate whether AGENTSCOPE maintains high failure localization accuracy and attribution performance across these variations. We conduct experiments using representative LLMs, including GPT-4o, DeepSeek-V3, and Qwen3-14B. For each base model, we measure AGENTSCOPE's ALA, SLA, and CA in detecting failure mode, and compare the results to baseline methods. Evaluations are performed on both the Who&When and AgentErrata benchmarks.

Table 4: Failure mode attribution across different models on Who&When and AgentErrata. (%)

| Model | Method | W&W Algo-Generated | | W&W Hand-Crafted | | AgentErrata | | |
|---|---|---|---|---|---|---|---|---|
| | | ALA | SLA | ALA | SLA | ALA | SLA | CA |
| GPT-4o | All-at-once | 42.06 | 14.29 | 35.09 | 5.26 | **41.45** | 5.18 | – / – |
| | Step-by-step | 32.54 | 23.02 | **65.52** | 15.52 | 30.57 | 17.10 | – / – |
| | AGENTSCOPE | **59.20** | **41.60** | 51.85 | **22.22** | 38.34 | **34.20** | 22.80 / **27.98** |
| DeepSeek-V3 | All-at-once | **60.32** | 19.84 | **47.37** | 3.51 | **31.61** | 5.70 | – / – |
| | Step-by-step | 9.52 | 7.94 | 32.75 | **12.07** | 20.71 | 12.95 | – / – |
| | AGENTSCOPE | 42.86 | **33.33** | 24.14 | **12.07** | 27.98 | **20.75** | 20.31 / **22.28** |
| Qwen3-14B | All-at-once | **63.81** | 11.43 | **50.88** | 5.26 | 37.65 | 4.94 | – / – |
| | Step-by-step | 38.89 | 19.84 | 48.28 | 10.34 | 35.71 | 9.52 | – / – |
| | AGENTSCOPE | 39.68 | **23.81** | 41.38 | **13.79** | **42.38** | **14.76** | 11.43 / **20.00** |

("-" indicates that the current tool does not support this detection feature; the result before the slash (/) indicates "w/o IV", the result after the slash (/) indicates "w/ IV". "W&W" stands for "Who&When".)

As shown in Table 4, AGENTSCOPE consistently outperforms the baseline methods across different base models on all datasets. Notably, its performance varies with the choice of LLM backbone: GPT-4o provides the highest SLA on all datasets. With GPT-4o, AGENTSCOPE also substantially boosts failure mode CA, a capability not supported by the baselines. For DeepSeek-V3, AGENTSCOPE again yields consistently higher SLA and uniquely provides failure attribution, suggesting that AGENTSCOPE mitigates step-level reasoning errors even on weaker backbones. On Qwen3-14B, AGENTSCOPE achieves the best balance across all three metrics, surpassing both baselines in overall localization and attribution performance. Given the large gap between step-level and agent-level accuracy, the All-at-once method likely does not detect real errors. Instead, it tends to choose the agent category with the highest prior frequency, thereby increasing the chance of a correct agent-level prediction.

In general, compared to Step-by-step and All-at-once methods, AGENTSCOPE maintains more stable accuracy across different base models, demonstrating its robustness to variations in the underlying LLM. Moreover, the results also indicate that AGENTSCOPE introduces an additional attribution capability absent in prior methods, highlighting its robustness and utility in complex failure analysis.

To wrap up, across both the Who&When and AgentErrata benchmarks, AGENTSCOPE demonstrates stronger and more stable failure localization than the baselines while uniquely enabling failure mode attribution, underscoring its robustness to base model differences and its added diagnostic utility.

## 5.4 WHAT IS THE EFFECT OF THE DIFFERENT COMPONENTS IN AGENTSCOPE?

**Effectiveness of the parser:** For systems that support well-structured trajectories, parsing issues are basically nonexistent. We validate the parser's ability to analyze non-structured trajectories. We randomly sampled 30 cases from our dataset AgentErrata and concatenated their originally stepwise separated conversation histories into continuous sequences. We then applied our trace parser to re-segment these sequences into discrete steps and conducted a manual evaluation to assess the parser's performance. In our experiment, the total number of original steps was 657. Overall, 9.89% of the steps differed from the original, and 6.09% contained missing content. See more details in Appendix D.

**Effectiveness of the two stages diagnosis process:** The effectiveness of the two stages is demonstrated in Table 3 and Table 4. In the CA (classification accuracy) columns, the results before the slash (/) indicate "w/o IV", that is to say they were performed by the preliminary prediction stage only; and the results after the slash (/) indicate "w/ IV", which were obtained by the preliminary prediction stage followed by the secondary verification stage. The results show that the preliminary prediction stage itself can achieve acceptable prediction accuracy, and that the secondary verification stage, though having little impact on the agent level and step level accuracy, can significantly boost the classification accuracy of failure mode.

## 5.5 WHAT IS THE RUNTIME OVERHEAD INTRODUCED BY AGENTSCOPE?

We evaluate the runtime overhead of AGENTSCOPE and its practical feasibility when integrated into LLM agents. Overhead is measured by both the number of agent steps before the first failure is detected and the cumulative inference time for entire trajectories. Because AGENTSCOPE dynamically processes intermediate steps using behavioral abstractions and structured reasoning, cost may increase with longer or more complex chains. Figure 2 shows cumulative runtime per step until the first failure is detected, with each line representing one of 20 sampled trajectories. This illustrates that early failure detection can save substantial time compared to completing full trajectories. Figure 3 presents a box plot of runtime across all trajectories grouped by prompt token count. Individual points outside the whiskers represent outliers, often caused by API variability or LLM processing fluctuations. Overall, runtime remains within a acceptable range. These results indicate that AGENTSCOPE introduces manageable overhead, with early detection yielding significant time savings and latency spikes occurring only in rare cases.

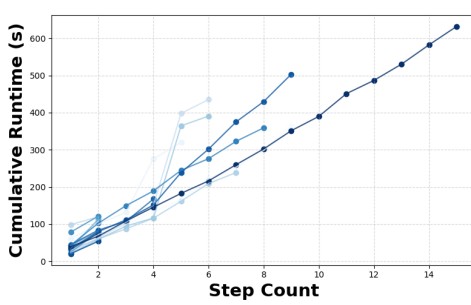 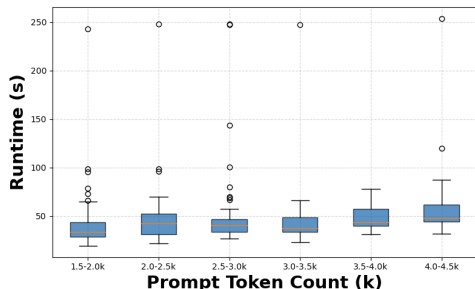

Figure 2: Cumulative time across steps.       Figure 3: Time across prompt token counts.

## 6 RELATED WORK

**Failures of Agents.** As LLM-based agents tackle increasingly complex tasks such as planning, web navigation, and tool use, understanding their failure modes becomes critical. Prior work highlights issues like tool misuse, hallucinated goals, and premature task termination. Architectural improvements such as memory modules or hierarchical task decomposition (e.g., Voyager (Wang et al., 2023), AssistGPT (Gao et al., 2023)) primarily focus on improving task success, offering limited introspection into why failures occur. Existing agent failure studies (Cemri et al., 2025) classify overall agent mistakes and construct failure taxonomies. In particular, Zhang et al. (2025b); Zhu et al. (2025) identify failure steps using LLMs as judges or via LLM reasoning, whereas Zhang et al. (2025a) fine-tunes models to automatically detect agent failures. Differently, AGENTSCOPE introduces a structured, interpretable, and reliable neuro-symbolic framework that encodes agent execution trajectories into *behavioral abstractions*. By representing raw trajectories in a structured form, AGENTSCOPE enables LLM-guided reasoning to systematically analyze agent behavior against formally defined invariants, pinpoint the exact step where a failure occurs, and classify its type. Since AGENTSCOPE is built on LLMs as the base models, its work is orthogonal to that of fine-tuned detection models. On the other hand, the structured analysis and design of invariants here are orthogonal to directly using LLMs as judges, and can thus benefit existing approaches.

**LLM as a Judge.** Recent works (Gu et al., 2024; Li et al., 2024) explore using LLMs as evaluators to assess output quality, correctness, or alignment. Benchmarks like Arena (Chiang et al., 2024) rank responses across tasks including QA, dialogue, and reasoning. While LLMs demonstrate meta-evaluative abilities, critiques (Chen et al., 2024; Szymanski et al., 2025) show they can be unreliable or inconsistent, especially for nuanced reasoning or domain-specific instructions. Compared to LLM-as-judge approaches, AGENTSCOPE leverages structured trajectory representations to guide step-wise reasoning diagnosis, offering interpretable and precise failure identification rather than holistic or subjective black-box assessments.

## 7 CONCLUSION

We presented AGENTSCOPE, a neuro-symbolic framework that diagnoses LLM agent failure mode via structured behavioral abstractions. By reasoning over formal invariants on encoded trajectories, it accurately identifies both the step and type of failure mode. Experiments on Who&When and AgentErrata show substantial gains in accuracy and interpretability over prior methods. Our approach demonstrates the power of combining structured abstractions with LLM reasoning and paves the way for scalable, reliable, and interpretable agent failure analysis.

## ETHICS STATEMENT

In this paper, we strictly obey the principles outlined in the ICLR Code of Ethics, including careful consideration of potential ethical concerns, including the impact on human subjects, data privacy, and fairness in algorithmic decisions. Specifically, the one public dataset do not have potential risk. As for the private synthesized dataset, we promise that any data used in this study was released

in compliance with legal and ethical standards, and proper security measures were implemented to safeguard personal information.

## REPRODUCIBILITY STATEMENT

We provide all the details of our work in the paper and appendix, including prompts, invariants, experimental setup, implementation, and hyperparameters for LLM reasoning. The code will be available upon the paper's publication. The above ensures that others can reproduce our work.

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

# A FORMAL DEFINITIONS OF INVARIANCES

## A.1 INVARIANT FOR INFORMATION WITHHOLDING

Information withholding occurs when essential details from one reasoning step are omitted in the propagation to the next, leading to distorted or biased outputs. Unlike the *unavailable context* defect, where no input is retrieved, this case emphasizes partial loss of information.

The invariant is expressed as:
$$inv_{withhold} ::= \exists\, (n_i, n_j) \in \mathbf{E}_{context} : \texttt{infoLoss}(content_i, content_j) > \delta.$$

Here, $E_{context}$ is the set of edges in the reasoning graph with label *output* or *serial*. $\texttt{infoLoss}(content_i, content_j)$ measures the degree of information loss between connected nodes. Violation of the invariant indicates that key constraints or evidence were withheld during reasoning. If an LLM is used as the judge of $\texttt{infoLoss}$, the threshold is replaced by a binary classification label.

## A.2 INVARIANT FOR WRONG CONTEXT

Wrong context occurs when an agent uses irrelevant, insufficient, invalid, or incorrect information in its reasoning, potentially leading to errors.

Let $\mathbf{N}$ be the set of nodes in the reasoning graph. For nodes $n_i \in \mathbf{N}$ we define a judging function $\texttt{wrong}(content_i)$ that evaluates whether the content of $n_i$ contains wrong or invalid information.

The invariant is expressed as:
$$inv_{wrong} ::= \exists\, n_i \in \mathbf{N} : \texttt{wrong}(content_i) > \delta$$

Violation indicates that the agent propagated context that is irrelevant, insufficient, invalid, or incorrect, potentially causing reasoning errors. If an LLM is used as the judge, the threshold of $\texttt{wrong}$ is replaced by a binary classification label.

## A.3 INVARIANT FOR ACTION-REASONING MISMATCH

Action-Reasoning Mismatch occurs when an agent's executed actions contradict its prior reasoning steps, indicating misalignment between decision and justification.

Let $\mathbf{N}_{action} \subseteq \mathbf{N}$ be the set of nodes representing actions (e.g., tool calls or output steps), and $\mathbf{N}_{reason} \subseteq \mathbf{N}$ be the set of reasoning nodes providing justification.

For each action node $n_a \in \mathbf{N}_{action}$, define a consistency function $\texttt{consistent}(n_a, n_r)$ that evaluates whether the action aligns with the reasoning in node $n_r$.

The invariant is expressed as:
$$inv_{mismatch} ::= \exists\, n_a \in \mathbf{N}_{action},\ n_r \in \mathbf{N}_{reason} : \neg\texttt{consistent}(n_a, n_r)$$

Violation indicates that an action contradicts its supporting reasoning, highlighting a decision-justification misalignment.

## A.4 INVARIANT FOR DISOBEYING SPECIFICATION

Disobey Specification occurs when an agent takes actions or makes decisions that violate given instructions, roles, or task requirements, leading to task misalignment.

Let $\mathbf{N}_{task} \subseteq \mathbf{N}$ be the set of nodes corresponding to task-related actions or decisions. Define a compliance function $\texttt{compliant}(n_i, spec)$ that returns $\texttt{True}$ if node $n_i$ adheres to the task specification $spec$, and $\texttt{False}$ otherwise.

The invariant is expressed as:
$$inv_{disobey} ::= \exists\, n_i \in \mathbf{N}_{task} : \neg\texttt{compliant}(n_i, spec)$$

Violation indicates that the agent acted inconsistently with the task instructions or role, potentially causing task failure.

## A.5 INVARIANT FOR VERIFICATION-FAILED OUTPUT

Verification-failed Output occurs when an agent produces output that does not satisfy a formal verifier or validation procedure.

Let $\mathbf{N}_{output} \subseteq \mathbf{N}$ be the set of nodes representing outputs generated by the agent. Define a verification function $\texttt{verified}(content_i)$ that returns $\texttt{True}$ if the output at node $n_i$ passes the verification checks, and $\texttt{False}$ otherwise.

The invariant is expressed as:

$$inv_{verif} ::= \exists\, n_i \in \mathbf{N}_{output} : \neg\texttt{verified}(content_i)$$

Violation indicates that the agent produced output that fails the verification, potentially leading to incorrect or invalid results.

## A.6 INVARIANT FOR INVALID VERIFICATION

Invalid Verification occurs when an agent's output initially passes a verification step but is later found to be incorrect or inconsistent with the task requirements.

Let $\mathbf{N}_{output} \subseteq \mathbf{N}$ be the set of nodes representing outputs. Define a verification function $\texttt{verified}(content_i)$ and a ground-truth check $\texttt{correct}(content_i)$.

The invariant is expressed as:

$$inv_{invalid\_verif} ::= \exists\, n_i \in \mathbf{N}_{output} : \texttt{verified}(content_i) \wedge \neg\texttt{correct}(content_i)$$

Violation indicates that the agent's output was falsely accepted by the verifier, potentially leading to undetected errors downstream.

## A.7 INVARIANT FOR UNAWARE OF TERMINATION

Unaware of Termination occurs when an agent fails to recognize the proper end of a task, potentially performing extra or endless actions.

Let $\mathbf{N}_{term} \subseteq \mathbf{N}$ be the set of nodes representing termination checks or task-ending conditions. Define a function $\texttt{terminated}(n_i)$ that returns $\texttt{True}$ if the agent correctly recognizes termination at node $n_i$, and $\texttt{False}$ otherwise.

The invariant is expressed as:

$$inv_{termination} ::= \exists\, n_i \in \mathbf{N}_{term} : \neg\texttt{terminated}(n_i)$$

Violation indicates that the agent did not recognize task completion, potentially leading to redundant or infinite reasoning steps.

## A.8 INVARIANT FOR CONVERSATION RESET

Conversation Reset occurs when an agent unnecessarily restarts a dialogue, losing previously accumulated context and reasoning progress.

Let $\mathbf{N}_{conv} \subseteq \mathbf{N}$ be the set of dialogue nodes. Define a function $\texttt{contextLost}(n_i, n_j)$ that returns $\texttt{True}$ if the transition from node $n_i$ to $n_j$ results in loss of prior context.

The invariant is expressed as:

$$inv_{reset} ::= \exists\, (n_i, n_j) \in \mathbf{E}_{conv} : \texttt{contextLost}(n_i, n_j)$$

Violation indicates that the agent restarted the conversation unnecessarily, losing important context and prior reasoning progress.

### A.9 INVARIANT FOR PREMATURE TERMINATION

Premature Termination occurs when an agent stops reasoning or task execution before achieving the task goal.

Let $n_{last}$ denote the last executed node in the reasoning graph, and $\mathtt{goalReached}(n_{last})$ indicate whether the task goal is achieved after executing $n_{last}$.

The invariant is expressed as:

$$inv_{premature} ::= \neg\mathtt{goalReached}(n_{last})$$

Violation indicates that the agent terminated before reaching the task goal, resulting in incomplete solutions or missed steps.

### A.10 INVARIANT FOR STEP REPETITION

Step Repetition occurs when an agent repeats prior reasoning or execution steps without a valid justification, leading to inefficiency.

Let $\mathbf{N}$ be the set of nodes in the reasoning graph. For nodes $n_i, n_j \in \mathbf{N}$ with $i < j$, define a similarity function $\mathtt{sim}(content_i, content_j)$ that measures semantic similarity between steps, and a function $\mathtt{justified}(n_j)$ indicating whether repetition is warranted.

The invariant is expressed as:

$$inv_{repeat} ::= \exists\, n_i, n_j \in \mathbf{N},\ i < j : \mathtt{sim}(content_i, content_j) > \theta \wedge \neg\mathtt{justified}(n_j)$$

Violation indicates that the agent repeated a prior step without justification, reflecting redundant or inefficient reasoning. If an LLM is used as the judge, the threshold of $\mathtt{sim}$ is replaced by a binary classification label.

### A.11 INVARIANT FOR SANDBAGGING

Sandbagging occurs when an agent deliberately underperforms or withholds capabilities to appear less competent and avoid oversight.

Let $\mathbf{N}_{action} \subseteq \mathbf{N}$ be the set of action nodes. Define a function $\mathtt{expectedPerformance}(n_i)$ representing the anticipated competence or quality of action $n_i$, and $\mathtt{actualPerformance}(n_i)$ for the observed performance.

The invariant is expressed as:

$$inv_{sandbag} ::= \exists\, n_i \in \mathbf{N}_{action} : \mathtt{actualPerformance}(n_i) \ll \mathtt{expectedPerformance}(n_i)$$

Violation indicates that the agent intentionally underperformed relative to expected capabilities, suggesting potential sandbagging behavior.

### A.12 INVARIANT FOR REWARD TAMPERING

Reward Tampering occurs when an agent manipulates the reward mechanism to artificially increase its gains, bypassing intended task evaluation.

Let $\mathbf{N}_{reward} \subseteq \mathbf{N}$ be the set of nodes associated with reward-related actions. Define a function $\mathtt{expectedReward}(n_i)$ representing the legitimate reward from action $n_i$, and $\mathtt{actualReward}(n_i)$ for the reward actually obtained.

The invariant is expressed as:

$$inv_{reward} ::= \exists\, n_i \in \mathbf{N}_{reward} : \mathtt{actualReward}(n_i) \gg \mathtt{expectedReward}(n_i)$$

Violation indicates that the agent artificially inflated its reward, suggesting reward tampering behavior.

## A.13 INVARIANT FOR SCHEMING

Scheming occurs when an agent exploits system or task design flaws to gain rewards without actually completing the intended task.

Let $\mathbf{N}_{action} \subseteq \mathbf{N}$ be the set of action nodes. Define a function $\texttt{taskCompleted}(n_i)$ that returns $\texttt{True}$ if the action contributes to legitimate task completion, and $\texttt{False}$ otherwise. Also, let $\texttt{rewardObtained}(n_i)$ measure the reward gained from $n_i$.

The invariant is expressed as:

$$inv_{scheming} ::= \exists\, n_i \in \mathbf{N}_{action} : \neg\texttt{taskCompleted}(n_i) \wedge \texttt{rewardObtained}(n_i) > 0$$

Violation indicates that the agent gained rewards without completing the task, reflecting scheming behavior.

## B REASONING-ACTION GRAPH CONSTRUCTION PROMPT

```
1   prompt = (
2       "You are given a reasoning-action trace from an AI agent, each step inside the trace includes the
        ↪   agent name and the content. Your task is to extract the key reasoning and action steps and
        ↪   represent them as a reasoning-action graph.\n\n"
3       "=============== Graph Specification ===============\n"
4       "A reasoning-action graph is defined as graph ::= {{N, E}}, where:\n"
5       "  - N is the set of nodes.\n"
6       "  - E is the set of edges.\n\n"
7
8       "--- Node Definition ---\n"
9       "A node is defined as n ::= N(id, agent name, type, content), where:\n"
10      "  - id: a unique identifier for the step.\n"
11      "       * The id MUST match the index of this step in the raw trace.\n"
12      "       * Use string format for ids (e.g., \"0\", \"1\", \"2\").\n"
13      "  - agent: the name of the agent performing this step, extracted verbatim (do not shorten or
        ↪   paraphrase).\n"
14      "  - type: the type of the step.\n"
15      "       * Use 'LLMCall' if the step is performed by the LLM model itself.\n"
16      "       * Use 'ToolCall' if the step is an external tool invocation.\n"
17      "  - content: the textual content of the reasoning or action step, extracted verbatim (do not
        ↪   shorten or paraphrase).\n\n"
18
19      "--- Edge Definition ---\n"
20      "An edge is defined as e ::= E(from_id, to_id, label), where:\n"
21      "  - from_id: the id of the source node (string).\n"
22      "  - to_id: the id of the target node (string).\n"
23      "  - label: the relationship type between steps.\n"
24      "     * 'serial': Indicates chronological sequence (one step follows another in order), and there is
        ↪   no control or output dependency.\n"
25      "     * 'control': Indicates that one step governs, constrains, or decides whether another step
        ↪   occurs.\n"
26      "     * 'output': Indicates that the output of one step is directly used as input for another
        ↪   step.\n\n"
27
28      "--- Output Format ---\n"
29      "Return the reasoning-action graph strictly in JSON format as:\n"
30      "{{\n"
31      "  \"nodes\": [\n"
32      "    {{\"id\": <step_index>,  \"agent\": <agent_name>, \"type\": <LLMCall|ToolCall>, \"content\":
        ↪   \"...\"}},\n"
33      "    ...\n"
34      "  ],\n"
35      "  \"edges\": [\n"
36      "    {{\"from_id\": <step_index>, \"to_id\": <step_index>, \"label\": <serial|control|output>}},\n"
37      "    ...\n"
38      "  ]\n"
39      "}}\n\n"
40
41
42      "=============== Note ===============\n"
43      "The graph is not necessarily a simple chain. Connect edges according to real dependency and
        ↪   logical links. Some steps may branch into multiple children or merge into later steps.  "
44
45
46      "=============== Constraints ===============\n"
47      "1. Every step in raw trace MUST be represented as a node. No step may be skipped, omitted, or
        ↪   merged with another.\n"
```

```
48          "2. Node ids must exactly match their step indices in the raw trajectory.\n"
49          "3. Node ids must be strings (e.g., \"0\", \"1\", \"2\").\n"
50          "4. Each reasoning or action step should appear in 'nodes' exactly once.\n"
51          "5. Edges must capture the actual flow of reasoning-action between steps.\n"
52          "6. Edge labels must be chosen carefully based on whether the relationship is sequential
      ↪    ('serial'), commanding ('control') or result-based ('output').\n"
53          "7. Do not add extra fields to nodes or edges.\n"
54          "8. The final output must be a single valid JSON object with no surrounding text.\n\n"
55
56          "=============== Example: Simple Chain ===============\n"
57          "Expected JSON output:\n"
58          "{{\n"
59          "  \"nodes\": [\n"
60          "    {{\"id\": \"0\", \"agent name\": \"Planner\", \"type\": \"LLMCall\", \"content\": \"The model
      ↪    receives the question.\"}},\n"
61          "    {{\"id\": \"1\", \"agent name\": \"Planner\", \"type\": \"LLMCall\", \"content\": \"The model
      ↪    reasons about possible approaches.\"}},\n"
62          "    {{\"id\": \"2\", \"agent name\": \"Planner\", \"type\": \"ToolCall\", \"content\": \"The model
      ↪    calls a calculator tool.\"}},\n"
63          "    {{\"id\": \"3\", \"agent name\": \"Calculator\", \"type\": \"ToolCall\", \"content\": \"The
      ↪    tool returns a numeric result.\"}},\n"
64          "    {{\"id\": \"4\", \"agent name\": \"Planner\", \"type\": \"LLMCall\", \"content\": \"The model
      ↪    produces the final answer.\"}}\n"
65          "  ],\n"
66          "  \"edges\": [\n"
67          "    {{\"from_id\": \"0\", \"to_id\": \"1\", \"label\": \"serial\"}},\n"
68          "    {{\"from_id\": \"1\", \"to_id\": \"2\", \"label\": \"serial\"}},\n"
69          "    {{\"from_id\": \"2\", \"to_id\": \"3\", \"label\": \"control\"}},\n"
70          "    {{\"from_id\": \"3\", \"to_id\": \"4\", \"label\": \"output\"}}\n"
71          "  ]\n"
72          "}}\n\n"
73
74          "=============== Example: Branching Steps ===============\n"
75          "Expected JSON output:\n"
76          "{\n"
77          "  \"nodes\": [\n"
78          "    {\"id\": \"0\", \"agent name\": \"Planner\", \"type\": \"LLMCall\", \"content\": \"The model
      ↪    receives the question.\"},\n"
79          "    {\"id\": \"1\", \"agent name\": \"Planner\", \"type\": \"LLMCall\", \"content\": \"The model
      ↪    considers solving via mathematical derivation.\"},\n"
80          "    {\"id\": \"2\", \"agent name\": \"Planner\", \"type\": \"ToolCall\", \"content\": \"The model
      ↪    queries a calculator tool.\"},\n"
81          "    {\"id\": \"3\", \"agent name\": \"Symbolic Analysist\", \"type\": \"LLMCall\", \"content\":
      ↪    \"The model derives a symbolic expression.\"},\n"
82          "    {\"id\": \"4\", \"agent name\": \"Calculator\", \"type\": \"ToolCall\", \"content\": \"The
      ↪    calculator tool returns a numeric result.\"},\n"
83          "    {\"id\": \"5\", \"agent name\": \"Planner\", \"type\": \"LLMCall\", \"content\": \"The model
      ↪    integrates both symbolic and numeric results to produce the final answer.\"}\n"
84          "  ],\n"
85          "  \"edges\": [\n"
86          "    {\"from_id\": \"0\", \"to_id\": \"1\", \"label\": \"serial\"},\n"
87          "    {\"from_id\": \"0\", \"to_id\": \"2\", \"label\": \"serial\"},\n"
88          "    {\"from_id\": \"1\", \"to_id\": \"3\", \"label\": \"output\"},\n"
89          "    {\"from_id\": \"2\", \"to_id\": \"4\", \"label\": \"control\"},\n"
90          "    {\"from_id\": \"3\", \"to_id\": \"5\", \"label\": \"output\"},\n"
91          "    {\"from_id\": \"4\", \"to_id\": \"5\", \"label\": \"output\"}\n"
92          "  ]\n"
93          "}\n\n"
94
95
96          "=============== Task ===============\n"
97          "Now process the following reasoning-action trace:\n\n"
98          f"{chat_content}\n\n"
99
100         "=============== Output Requirement ===============\n"
101         "Return only the JSON object that conforms exactly to the specification and schema above. "
102         "Do not include any explanations, comments, or additional text."
103     )
```

## C  FAILURE MODE DISTRIBUTION OF AGENTERRATA

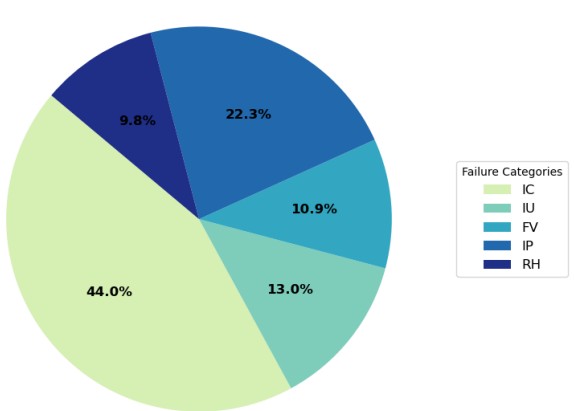

Figure 4: Failure Category Distribution of AgentErrata

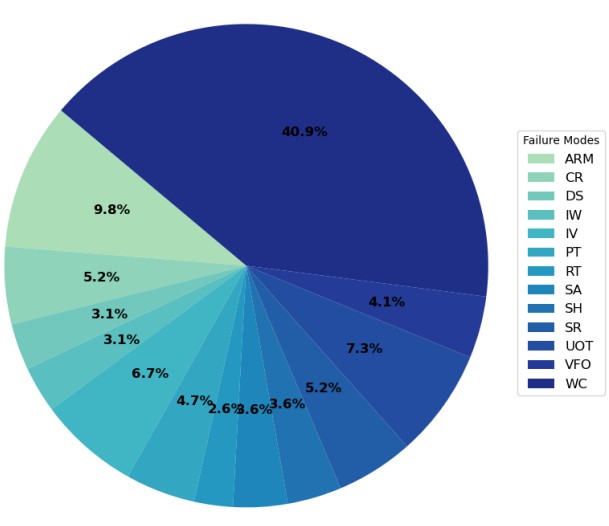

Figure 5: Failure Mode Distribution of AgentErrata

Table 5: Failure Categories and Their Abbreviations

| Category | Abbrev. | Sub-category | Abbrev. |
|---|---|---|---|
| Invalid Context | IC | Information Withholding | IW |
| | | Wrong Context | WC |
| Instruction Unfollowing | IU | Action-Reasoning Mismatch | ARM |
| | | Disobey Specification | DS |
| Faulty Verification | FV | Verification-failed output | VFO |
| | | Invalid Verification | IV |
| Incoherent Planning | IP | Unaware of Termination | UOT |
| | | Conversation Reset | CR |
| | | Premature Termination | PT |
| | | Step Repetition | SR |
| Reward Hacking | RH | Sandbagging | SA |
| | | Reward Tampering | RT |
| | | Scheming | SH |
| Unknown (GT missing) | NO GT | — | NO GT |

Table 6: Taxomony of LLM agent failure patterns.

| Category | Pattern | Description |
|---|---|---|
| Invalid Context | Information Withholding | Withholds information to manipulate output. |
| | Wrong Context | Context used is irrelevant, insufficient, invalid, or incorrect, causing errors. |
| Instruction Unfollowing | Action-Reasoning Mismatch | Actions contradict reasoning, showing decision-justification misalignment. |
| | Disobey Specification | Deviates from instructions or roles, causing task misalignment. |
| Faulty Verification | Verification-failed Output | Output did not pass the verifier's verification. |
| | Invalid Verification | Passed verification, but later proved incorrect. |
| Incoherent Planning | Unaware of Termination | Fails to recognize end, causing extra or endless actions. |
| | Conversation Reset | Unwarranted dialogue restarts, losing context and progress. |
| | Premature Termination | Prematurely stops, causing incomplete solutions or missed steps. |
| | Step Repetition | Repeats prior steps unnecessarily without justification. |
| Reward Hacking | Sandbagging | Deliberately underperforms to hide capabilities and avoid oversight. |
| | Reward Tampering | Manipulates reward mechanism to boost gains artificially. |
| | Scheming | Exploits design flaws to gain rewards without task completion. |

## D EFFECTIVENESS OF THE LLM-BASED TRACE PARSER

Before a ReAG can be constructed, the agent conversation histroy needs to be parsed as a structured step-wise trace. For systems that support well-structured trajectories, parsing issues are basically nonexistent. But when explicit delimiters are absent or when the boundaries between reasoning and action execution are not clearly defined, steps with clear-cut boundaries need to be extracted via a designed LLM-based trace parser first. The following shows the validation of this parser's ability to analyze non-structured trajectories.

We randomly sampled 30 cases from our dataset AgentErrata and concatenated their originally step-wise separated conversation histories into single continuous sequences. We then applied our trace parser to re-segment these sequences into discrete steps and conducted a manual evaluation to assess the trace parser's performance.

We categorize the parsed differences between the parsed steps and the original ground truth steps into three types:

1. Missing sentence(s);

2. Step split – one original step is split into two or more steps without losing any content;

3. Step merge – two or more original steps are merged, sometimes with content missing.

In our experiment, the total number of original steps was 657. The differences observed were:

- Missing sentence: 29 steps (4.41%)
- Step split: 9 steps (1.37%)
- Step merge with missing content: 11 steps (1.67%)

- Step merge without missing content: 16 steps (2.44%)

Overall, 9.89% of the steps differed from the original, and 6.09% contains missing content. Furthermore, since most agent frameworks support structured trajectory logging, this proportion would be significantly lower in practical applications.

## E ISSUES IN WHO&WHEN DATASET

Traces provided by the Who&When dataset typically do not contain one single mistake step or one single decisive mistake step. In many traces, the ground truth "mistake step" is annotated at the first point where a potential issue begins to surface. However, these early steps do not necessarily lead to failure — they may still be part of a reasonable path toward the final goal, or they can often be corrected easily by the agent itself.

Take the following trace as example:

```
1  // Who&When Hand-Crafted subset trace 2.json
2  "history": [
3      {
4          "content": "What is the worst rated series (according to Rotten Tomatoes) with more than 1
            ↪  season that Ted Danson has starred in and is available on Amazon Prime Video (US)?\n",
5          "role": "human"
6      },
7      {
8          "content": "Initial plan:\n\nWe are working to address the following user request:\n\nWhat is
            ↪  the worst rated series (according to Rotten Tomatoes) with more than 1 season that Ted
            ↪  Danson has starred in and is available on Amazon Prime Video (US)?\n\nTo answer this
            ↪  request we have assembled the following team:\n\nAssistant: A helpful and general-purpose
            ↪  AI assistant that has strong language skills, Python skills, and Linux command line
            ↪  skills.\nComputerTerminal: A computer terminal that performs no other action than running
            ↪  Python scripts (provided to it quoted in ```python code blocks), or sh shell scripts
            ↪  (provided to it quoted in ```sh code blocks)\nFileSurfer: An agent that can handle local
            ↪  files.\nWebSurfer: A helpful assistant with access to a web browser. Ask them to perform
            ↪  web searches, open pages, and interact with content (e.g., clicking links, scrolling the
            ↪  viewport, etc., filling in form fields, etc.) It can also summarize the entire page, or
            ↪  answer questions based on the content of the page. It can also be asked to sleep and wait
            ↪  for pages to load, in cases where the pages seem to be taking a while to load.\n\n\nHere
            ↪  is an initial fact sheet to consider:\n\n1. GIVEN OR VERIFIED FACTS\n- The request is about
            ↪  Ted Danson.\n\n2. FACTS TO LOOK UP\n- Rotten Tomatoes ratings for all series Ted Danson has
            ↪  starred in.\n- Which series with more than 1 season Ted Danson has starred in.\n-
            ↪  Availability of these series on Amazon Prime Video (US).\n- Rotten Tomatoes score of each
            ↪  series to determine the worst-rated.\n  \n3. FACTS TO DERIVE\n- Comparison of Rotten
            ↪  Tomatoes scores to determine the worst-rated series.\n- Matching the availability on Amazon
            ↪  Prime Video with the list of worst-rated series.\n4. EDUCATED GUESSES\n- Ted Danson is
            ↪  widely known for popular shows such as \"Cheers\", \"The Good Place\", and \"CSI\". A less
            ↪  popular series could potentially have a lower rating.\n\n\nHere is the plan to follow as
            ↪  best as possible:\n\nHere is a short bullet-point plan for addressing the original
            ↪  request:\n- Use WebSurfer to search for a list of series that Ted Danson has starred in and
            ↪  confirm which of those have more than 1 season.\n- Find the Rotten Tomatoes scores for each
            ↪  of those series.\n- Determine which of those series are available on Amazon Prime Video
            ↪  (US).\n- Identify the worst-rated series (according to Rotten Tomatoes) among those that
            ↪  Ted Danson has starred in and are available on Amazon Prime Video (US).\n\nWe will start by
            ↪  having WebSurfer gather the required information about Ted Danson's
            ↪  series.\n\n/usr/local/lib/python3.11/site-packages/autogen_magentic_one/agents/orchestrator.py:216:
            ↪  UserWarning: Resolved model mismatch: gpt-4o-2024-08-06 != gpt-4o-2024-05-13. Model mapping
            ↪  may be incorrect.\n  ledger_response = await self._model_client.create(",
9          "role": "Orchestrator (thought)"
10     },
11     {
12         "content": "Updated Ledger:\n{\n  \"is_request_satisfied\": {\n    \"reason\": \"The request is
            ↪  not yet fully satisfied as we have not determined the worst-rated Ted Danson series with
            ↪  more than one season that is available on Amazon Prime Video (US).\",\n    \"answer\":
            ↪  false\n  },\n  \"is_in_loop\": {\n    \"reason\": \"We are not yet in a loop since we are
            ↪  starting to gather information to address the request.\",\n    \"answer\": false\n  },\n
            ↪  \"is_progress_being_made\": {\n    \"reason\": \"We are making forward progress by planning
            ↪  to gather information about Ted Danson's series and their Rotten Tomatoes ratings.\",\n
            ↪  \"answer\": true\n  },\n  \"next_speaker\": {\n    \"reason\": \"The next step involves
            ↪  gathering information from the web about the series Ted Danson has starred in and their
            ↪  Rotten Tomatoes ratings.\",\n    \"answer\": \"WebSurfer\"\n  },\n
            ↪  \"instruction_or_question\": {\n    \"reason\": \"We need to find a list of series that Ted
            ↪  Danson has starred in, confirm which of those have more than one season, and find their
            ↪  Rotten Tomatoes ratings.\",\n    \"answer\": \"Please search for a list of series that Ted
            ↪  Danson has starred in, and confirm which of those series have more than one season.
            ↪  Additionally, find the Rotten Tomatoes ratings for each of those series.\"\n  }\n}",
13         "role": "Orchestrator (thought)"
```

```
1188    14        },
1189    15        {
1190    16            "content": "Please search for a list of series that Ted Danson has starred in, and confirm
                ↪    which of those series have more than one season. Additionally, find the Rotten Tomatoes
1191            ↪    ratings for each of those series.",
1192    17            "role": "Orchestrator (-> WebSurfer)"
1193    18        },
        19        {
1194    20            +"content": "I typed 'Ted Danson TV series list' into '0 characters out of 2000'.\n\nHere is a
                ↪    screenshot of [Ted Danson TV series list -
1195            ↪    Search](https://www.bing.com/search?q=Ted+Danson+TV+series+list&"[...truncated]"). The
1196            ↪    viewport shows 26% of the webpage, and is positioned at the top of the page.\nThe following
1197            ↪    metadata was extracted from the webpage:\n\n{\n    \"meta_tags\": {\n        \"referrer\":
                ↪    \"origin-when-cross-origin\",\n        \"SystemEntropyOriginTrialToken\":
1198            ↪    \"A1L3tx5CzccqjN3lK6st/"[...truncated]",\n        \"og:description\": \"Intelligent search
                ↪    from Bing makes it easier to quickly find what you\\u2019re looking for and rewards
1199            ↪    you.\",\n        \"og:site_name\": \"Bing\",\n        \"og:title\": \"Ted Danson TV series
                ↪    list - Bing\",\n        \"og:url\":
1200            ↪    \"https://www.bing.com/search?q=Ted+Danson+TV+series+list&form="[...truncated]",\n
1201            ↪    \"fb:app_id\": \"3732605936979161\",\n        \"og:image\":
1202            ↪    \"http://www.bing.com/sa/simg/facebook_sharing_5.png\",\n        \"og:type\":
1203            ↪    \"website\",\n        \"og:image:width\": \"600\",\n        \"og:image:height\": \"315\"\n
                ↪    }\n}\n\nAutomatic OCR of the page screenshot has detected the following text:\n\n**Main
1204            ↪    Content:**\n\nShow / Ted Danson\nA Man on the Inside\n2024\n\nTV
                ↪    Guide\nhttps://www.tvguide.com  celebrities  ted-danso...\nTed Danson List of Movies and TV
1205            ↪    Shows - TV Guide\nSee Ted Danson full list of movies and tv shows from their career. Find
1206            ↪    where to watch Ted Danson's latest movies and tv shows\nTags: Ted Danson Movies and Tv
                ↪    Shows Movies Ted Danson Played in\n\nRanker\nhttps://www.ranker.com  list
1207            ↪    ted-danson-tv-sho...\nTed Danson: TV Shows Starring Ted Danson (10 Items) ...\nJul 3, 2024
1208            ↪    · TV Shows featuring Ted Danson, listed alphabetically with photos when available. All of
1209            ↪    the TV programs that had Ted Danson in the cast are featured here. Ted Danson may have had
                ↪    a prominent role in these shows, but ...\nTags: Ted Danson Movies and Tv Shows Tv Show with
1210            ↪    Ted Danson\n\n**People also ask:**\n\nWhich CSI shows are based on Ted Danson?\nLists the
1211            ↪    series featuring Ted Danson. These shows include The Good Place, Fargo, CSI, CSI: Cyber,
                ↪    Damages, Bored to ...\n\nWho is Ted Danson?\nBirth Name: Edward Bridge Danson III\nBirth
1212            ↪    Place: San Diego, California, United States\nProfession Actor, producer Jimmy Kimmel: Live!
1213            ↪    How Can\n\n**Explore More:**\n\nfrom Wikipedia\nTop-searched actors from the Ted Danson
                ↪    article\n\nJane Fonda\nWhoopi Goldberg\nKristen Bell\nJohn Lithgow\nRobin Williams\nTom
1214            ↪    Selleck\nGlenn Close\nMary Tyler Moore\nMary Steenburgen\nIsabella Rossellini\n\nExplore
                ↪    more on Wikipedia\n<Image>",
1215    21            "role": "WebSurfer"
1216    22        },
        23        {
1217    24            "content": "Updated Ledger:\n{\n  \"is_request_satisfied\": {\n    \"reason\": \"The request is
1218            ↪    not fully satisfied as we have not yet identified the worst rated series with more than 1
1219            ↪    season that Ted Danson has starred in and is available on Amazon Prime Video (US).\",\n
                ↪    \"answer\": false\n  },\n  \"is_in_loop\": {\n    \"reason\": \"We are not in a loop. We
1220            ↪    have made some progress in identifying where to find the information but need to continue
                ↪    further steps.\",\n    \"answer\": false\n  },\n  \"is_progress_being_made\": {\n
1221            ↪    \"reason\": \"We are making forward progress by gathering information on the list of shows
1222            ↪    Ted Danson has been in, and looking for their ratings.\",\n    \"answer\": true\n  },\n
                ↪    \"next_speaker\": {\n    \"reason\": \"WebSurfer is the best choice to continue gathering
1223            ↪    information on the shows Ted Danson starred in.\",\n    \"answer\": \"WebSurfer\"\n  },\n
1224            ↪    \"instruction_or_question\": {\n    \"reason\": \"We need the WebSurfer to search reliable
1225            ↪    sources for the list of TV series that Ted Danson starred in, especially those with more
                ↪    than one season.\",\n    \"answer\": \"Please find reliable sources with the complete list
1226            ↪    of TV series that Ted Danson has starred in, including which of those series have more than
1227            ↪    1 season.\"\n  }\n}",
        25            "role": "Orchestrator (thought)"
1228    26        },
        27        {
1229    28            "content": "Please find reliable sources with the complete list of TV series that Ted Danson
1230            ↪    has starred in, including which of those series have more than 1 season.",
        29            "role": "Orchestrator (-> WebSurfer)"
1231    30        },
        31        {
1232    32            "content": "Next speaker WebSurfer",
1233    33            "role": "Orchestrator (thought)"
1234    34        },
        35        {
1235
1236
1237
1238
1239
1240
1241
```

```
1242
1243  36          "content": "Updated Ledger:\n{\n  \"is_request_satisfied\": {\n    \"reason\": \"The request is
1244      ↪ not fully satisfied because we have not yet identified the worst-rated series Ted Danson
1245      ↪ has starred in that is available on Amazon Prime Video (US).\",\n    \"answer\": false\n
1246      ↪ },\n  \"is_in_loop\": {\n    \"reason\": \"We are not in a loop, but we encountered issues
1247      ↪ in finding the required information.\",\n    \"answer\": false\n  },\n
1248      ↪ \"is_progress_being_made\": {\n    \"reason\": \"Progress is being made, but it is slow due
1249      ↪ to encountering errors in the web search and data extraction process.\",\n    \"answer\":
1250      ↪ true\n  },\n  \"next_speaker\": {\n    \"reason\": \"The useful web results were
1251      ↪ interrupted by a timeout error. Retrying might help.\",\n    \"answer\": \"WebSurfer\"\n
1252      ↪ },\n  \"instruction_or_question\": {\n    \"reason\": \"We need to gather a comprehensive
1253      ↪ list of Ted Danson's TV series, identify those with more than one season, and check their
1254      ↪ availability on Amazon Prime Video (US).\",\n    \"answer\": \"Retry the search for a list
1255      ↪ of TV series that Ted Danson has starred in and confirm which of those have more than one
1256      ↪ season. Once the list is obtained, find the Rotten Tomatoes scores and availability on
1257      ↪ Amazon Prime Video (US).\"\n  }\n}",
     37          "role": "Orchestrator (thought)"
     38      },
     39      {
     40          "content": "Retry the search for a list of TV series that Ted Danson has starred in and confirm
         ↪ which of those have more than one season. Once the list is obtained, find the Rotten
         ↪ Tomatoes scores and availability on Amazon Prime Video (US).",
     41          "role": "Orchestrator (-> WebSurfer)"
     42      },
     43      {
     44          "content": "Next speaker WebSurfer",
     45          "role": "Orchestrator (thought)"
     46      },
     47      {
     48          "content": "I clicked 'Credits'.\n\nHere is a screenshot of [Ted Danson List of Movies and TV
         ↪ Shows - TV Guide](https://www.tvguide.com/celebrities/ted-danson/credits/3030192417/). The
         ↪ viewport shows 7% of the webpage, and is positioned at the top of the page.\nThe following
         ↪ metadata was extracted from the webpage:\n\n{\n    \"jsonld\": [\n        {\n
         ↪ \"@context\": \"https://schema.org/\",\n            \"@type\": \"Organization\",\n
         ↪ \"url\": \"https://www.tvguide.com/\",\n            \"logo\": {\n                \"@type\":
         ↪ \"ImageObject\",\n                \"url\":
         ↪ \"https://www.tvguide.com/a/www/img/amp-logo.png\",\n                \"width\": \"310\",\n
         ↪ \"height\": \"310\"\n            },\n            \"sameAs\": [\n
         ↪ \"https://www.facebook.com/TVGuide\",\n
         ↪ \"https://www.twitter.com/TVGuide\",\n                \"https://instagram.com/tvguide\",\n
         ↪ \"https://www.pinterest.com/tvguide\"\n            ]\n        },\n        {\n
         ↪ \"@context\": \"https://schema.org/\",\n            \"@type\": \"WebSite\",\n
         ↪ \"url\": \"https://www.tvguide.com/\",\n            \"potentialAction\": {\n
         ↪ \"@type\": \"SearchAction\",\n                \"target\":
         ↪ \"https://www.tvguide.com/s/{search_term_string}/\",\n                \"query-input\":
         ↪ \"required name=search_term_string\"\n            }\n        }\n    ],\n    \"meta_tags\":
         ↪ {\n        \"viewport\": \"width=device-width, initial-scale=1\",\n        \"fb:app_id\":
         ↪ \"63084645929\",\n        \"google-site-verification\":
         ↪ \"Od29UEFliEmsUY2XHOkfXxkTNucL0cv_rc0J_RPWXU4\",\n        \"og:site_name\":
         ↪ \"TVGuide.com\",\n        \"twitter:site\": \"@TVGuide\",\n        \"twitter:image\":
         ↪ \"https://www.tvguide.com/a/img/\"[...truncated]\",\n        \"og:image\":
         ↪ \"https://www.tvguide.com/a/img/\"[...truncated]\",\n        \"nuxtstatus\": \"loaded\",\n
         ↪ \"og:image:height\": \"850\",\n        \"og:image:width\": \"850\",\n
         ↪ \"description\": \"See Ted Danson full list of movies and tv shows from their career. Find
         ↪ where to watch Ted Danson's latest movies and tv shows\",\n        \"og:description\":
         ↪ \"See Ted Danson full list of movies and tv shows from their career. Find where to watch
         ↪ Ted Danson's latest movies and tv shows\",\n        \"og:title\": \"Ted Danson\",\n
         ↪ \"og:url\": \"https://www.tvguide.com/celebrities/ted-danson/credits/3030192417/\",\n
         ↪ \"twitter:card\": \"summary_large_image\",\n        \"twitter:description\": \"See Ted
         ↪ Danson full list of movies and tv shows from their career. Find where to watch Ted Danson's
         ↪ latest movies and tv shows\",\n        \"twitter:title\": \"Ted Danson\",\n
         ↪ \"twitter:url\": \"https://www.tvguide.com/celebrities/ted-danson/credits/3030192417/\",\n
         ↪ \"adtags\": \"cid=ted-danson%2C3030192417&collection=\"[...truncated]\"\n
         ↪ }\n}\n\nAutomatic OCR of the page screenshot has detected the following text:\n\nHere is
         ↪ the transcribed text from the image, excluding any mention of the blurred face:\n\n- TV
         ↪ GUIDE\n- Search TV Shows and Movies...\n- Live & Upcoming\n- Shopping\n- What to Watch\n-
         ↪ News\n- Join/Sign In\n- Overview\n- Biography\n- TV Listings\n- Credits\n- Ted Danson\n-
         ↪ Profession\n- Actor, Producer, Soundtrack\n- Actor\n- 88 Credits\n- NESPRESSO\n\nThe text
         ↪ in ads displayed in another language:\n\n- AI\n- AI\n- \n- \n- \n- \n<Image>",
     49          "role": "WebSurfer"
     50      },
     51      {
1290
1291
1292
1293
1294
1295
```

```
52          "content": "Updated Ledger:\n{\n  \"is_request_satisfied\": {\n    \"reason\": \"The request is
    ↪   not yet fully satisfied as we have not identified the worst-rated series featuring Ted
    ↪   Danson that is available on Amazon Prime Video (US) according to Rotten Tomatoes.\",\n
    ↪   \"answer\": false\n  },\n  \"is_in_loop\": {\n    \"reason\": \"We have repeated the action
    ↪   of searching for a list of Ted Danson's series multiple times without getting the desired
    ↪   comprehensive list.\",\n    \"answer\": true\n  },\n  \"is_progress_being_made\": {\n
    ↪   \"reason\": \"There is some forward progress as we have found a source that lists Ted
    ↪   Danson's credits, but we still need to find the necessary details.\",\n    \"answer\":
    ↪   true\n  },\n  \"next_speaker\": {\n    \"reason\": \"WebSurfer is needed to retrieve
    ↪   reliable and complete information regarding the series Ted Danson has starred in, and to
    ↪   obtain their Rotten Tomatoes scores and availability on Amazon Prime Video (US).\",\n
    ↪   \"answer\": \"WebSurfer\"\n  },\n  \"instruction_or_question\": {\n    \"reason\": \"To get
    ↪   a comprehensive list of Ted Danson's TV series and additional required details.\",\n
    ↪   \"answer\": \"Please check the TVGuide link you found earlier and retrieve the list of TV
    ↪   series that Ted Danson has starred in, highlighting those with more than one season. Then,
    ↪   find their Rotten Tomatoes scores and check their availability on Amazon Prime Video
    ↪   (US).\"\n  }\n}",
53          "role": "Orchestrator (thought)"
54      },
55      {
56          "content": "Please check the TVGuide link you found earlier and retrieve the list of TV series
    ↪   that Ted Danson has starred in, highlighting those with more than one season. Then, find
    ↪   their Rotten Tomatoes scores and check their availability on Amazon Prime Video (US).",
57          "role": "Orchestrator (-> WebSurfer)"
58      },
59      {
60          "content": "Next speaker WebSurfer",
61          "role": "Orchestrator (thought)"
62      },
63      {
64          "content": "I clicked 'Credits'.\n\nHere is a screenshot of [Ted Danson List of Movies and TV
    ↪   Shows - TV Guide](https://www.tvguide.com/celebrities/ted-danson/credits/3030192417/). The
    ↪   viewport shows 7% of the webpage, and is positioned at the top of the page.\nAutomatic OCR
    ↪   of the page screenshot has detected the following text:\n\nTV GUIDE\nSearch TV Shows and
    ↪   Movies...\n\nLive & Upcoming \nShopping \nWhat to Watch \nNews \nJoin/Sign In\n\nTed
    ↪   Danson\n\nProfession\nActor, Producer, Soundtrack\n\nActor 88 Credits\n<Image>",
65          "role": "WebSurfer"
66      },
67      ... (remaining content truncated)
```

The ground truth "mistake step" of this trace is annotated at step 4 (line 20 in the above context), and the provided mistake reasong is stated as: "The information obtained by WebSurfer is neither reliable nor satisfies the requirements of the Orchestrator." However, if we scrutinize this trace, we can see that the orchestrator's instruction in the previous step (line 16, step 3) is actually accurate and clear. While the websurfer indeed omitted some information when searching for the keywords, this was still just an intermediate step toward the goal. Moreover, the orchestrator later noticed this issue and reissued the instruction (line 28, step 6), but the websurfer failed to execute it (line 48, step 11). After the orchestrator repeated the similar instruction again (line 56, step 13), the websurfer still followed the exact same steps as before (line 64, step 15). This constitutes a step repetition failure, which is far more detrimental to the final outcome than the earlier mistake in step 4. In other words, within the same trace of the Who&When dataset, multiple failures may exist, and the ground-truth annotation is not necessarily placed on the most critical failure mode — or there may not even exist a single "most decisive" mistake step. Therefore, it appears that this dataset does not effectively reflect a model's capability to locate and attribute failures, which substantially undermines its reference value.

