# OpenReview forum: "Diagnosing with Insights: Structured Analysis of Agent Failures via Behavioral Abstractions"
_ICLR.cc/2026/Conference — Submitted to ICLR 2026_

### Official Review · Reviewer_gdmo · 2025-10-25

**Soundness:** 3
**Presentation:** 3
**Contribution:** 3
**Rating:** 6
**Confidence:** 2

**Summary:**

This paper introduces AGENTSCOPE, a neuro-symbolic framework for diagnosing failures in LLM-based agent systems. The core contribution is a two-fold approach: first, abstracting complex, unstructured agent trajectories into a structured Reasoning-Action Graph (ReAG); and second, using LLM-guided reasoning to check this graph for violations of pre-defined "Neural Invariants." These invariants are derived from a comprehensive taxonomy of agent failures. The authors evaluate their method on the existing Who&When benchmark and a new, purpose-built benchmark called AgentErrata. The results demonstrate that AGENTSCOPE significantly outperforms baseline methods in both failure localization (pinpointing the error step) and failure attribution (classifying the error type), a capability that baselines lack.

**Strengths:**

1. The challenge of debugging and diagnosing failures in complex, multi-step LLM agent systems is a critical bottleneck for their reliable deployment. This paper addresses a highly relevant and practical problem for the community.

2. The core idea of combining a structured, symbolic representation (the ReAG) with the semantic reasoning capabilities of LLMs (to check Neural Invariants) is powerful. It provides a principled way to move beyond unreliable, holistic "LLM-as-a-judge" approaches by imposing structure on the diagnostic process.

3. This paper proposes a new benchmark called AgentErrata. The authors use the method of using failure-taxonomy-guided fault injection to create a diverse and well-annotated dataset for both localization and attribution. This will be a valuable asset for future research in agent reliability.

4. On AgentErrata, AGENTSCOPE improves SLA for attribution from 5.35% to 30.48% and CA from 4.40% to 19.05% (with IV). These are sizable improvements over LLM-as-judge.

5. AgentErrata spans a wide taxonomy (invalid context, instruction-unfollowing, faulty reasoning, incoherent planning, reward hacking), and the appendix lists categories and abbreviations.

**Weaknesses:**

1. The most significant weakness of the paper is that while the concept of Neural Invariants is a core contribution, the paper remains extremely vague about their actual implementation. The appendix provides formal definitions, but these rely on abstract functions like infoLoss(), sim(), relevant(), and isValid().

2. While AGENTSCOPE's relative improvement in Classification Accuracy (CA) is impressive, the absolute accuracy of 19.05% (with the Invariant Verifier) is still quite low for any practical application. This suggests that reliable failure attribution is an extremely challenging task, and while AGENTSCOPE is a step in the right direction, it is far from a complete solution.

3. The paper lacks any evaluation or ablation study on the robustness of the diagnosis pipeline to errors in the ReAG construction phase.

**Questions:**

1. The paper needs to clarify:
(1) Are these "neural functions" implemented as specific, engineered prompts to a general-purpose LLM?
(2) Are they fine-tuned classifiers?
(3) How are thresholds determined? Without these details, the method is not reproducible, and it is difficult for the reader to assess the true complexity and novelty of the invariant-checking mechanism versus simple prompting.

2. Regarding the two-stage diagnosis process: Could you further clarify the concrete difference between the invariant checks performed in the "preliminary prediction" stage versus the "secondary verification" stage? For instance, does the first stage use a simpler prompt or a smaller context window to improve efficiency?

3. Regarding ReAG construction: Have you performed any analysis on the accuracy of the LLM-based parser used to generate the ReAG from unstructured logs?

---

> ### Author Response · Authors · 2025-11-21
>
> Thank you for the detailed and constructive feedback! We treasure the opportunity to address your concerns and improve our work.
>
> ## Weakness 1: The most significant weakness of the paper is that while the concept of Neural Invariants is a core contribution, etc.
>
> This "weakness" issue will be combined with question 1 for a single reply.
>
> ## Weakness 2: While AGENTSCOPE's relative improvement in Classification Accuracy (CA) is impressive, etc.
>
> Thank you for the insightful comment. We acknowledge that the current results show room for improvement. We would like to clarify that some of the reported numbers were affected by issues in the earlier version of the dataset and evaluation pipeline. We are correcting these issues and are generating updated results.
> We will provide partial updated results first and a fully revised version of the paper with complete corrected numbers by Dec 3. Our preliminary findings indicate that the performance improves notably after fixing the issues. We appreciate the reviewer’s feedback and will ensure that the final revision presents more accurate and up-to-date results.
>
> Currently, we sampled 52 traces from AgentErrata and made a preliminary refinement, here are results:
>
> |data version|ALA | SLA | CA |
> |:-:|:-:|:-:|:-:|
> | original | 47.17% | 33.96% | 13.21% |
> | refined | 51.92% | 48.08% | 21.15% |
>
> (the experiment settings were: GPT-4o as base LLM model, without ground truth of the original problem, with Invariant Verifier)
>
> Due to time constraints, we are currently only testing on a small set. We believe that the full experiment will be run in the next revised version, and issues will be fixed with further improvements.
>
> ## Weakness 3: The paper lacks any evaluation or ablation study on the robustness of the diagnosis pipeline, etc.
>
> This "weakness" issue will be combined with question 3 for a single reply.

---

> ### Author Response · Authors · 2025-11-21
>
> ## Question 1:  The paper needs to clarify: (1) Are these "neural functions" implemented as specific, etc.
>
> Thank you for the question. Neural functions are implemented as task-specific prompts to the general-purpose LLM (e.g., GPT-4o) and are not fine-tuned classifiers. In the current design, the LLM directly makes judgments without using explicit thresholds.
>
> For example, the following shows the prompt of a neural function implementation (simNode()) for the failure mode “Step Repetition”.It is necessary to traverse adjacent nodes in the graph and apply the current function to each pair of nodes.
>
> prompt = (
>
>     "You are given two nodes from a reasoning execution graph representing an AI agent's problem-solving process. Judge whether the two nodes are semantically similar or not. Return only 'Yes' (for similar) or 'No' (for not similar). The nodes are considered similar if they convey the same meaning, perform the same function, or represent the same step in the reasoning process, even if they are worded differently. Conversely, if the nodes have different meanings, functions, or represent different steps, they are considered not similar. Even subtle similarities should lead to a 'Yes' answer.\n"
>     f"Node 1: {node1}\n"
>     f"Node 2: {node2}\n"
>     "Your response should be either 'Yes' or 'No' without any additional explanation."
>
> )
>
> Furthermore, since AgentScope supports a plugin-based implementation, users can optionally configure threshold-based criteria if desired.
> We will clarify this in the appendix in the revised version for reproducibility and clarity.
>
> ## Question 2: Regarding the two-stage diagnosis process, etc.
>
> Thank you for the question. The two-stage diagnosis process in AgentScope is designed to balance efficiency and accuracy:
>
> - In the preliminary prediction stage, invariant checks are performed over a large context window to quickly identify potential failure steps.
>
> - In the secondary verification stage (Invariant Verifier), the same invariants are re-evaluated using more precise prompts and considering the invariant patterns, to reduce false positives and confirm the initial predictions.
>
> This staged design ensures reliable failure localization while keeping the verification process efficient. We will clarify these implementation details in the revised appendix.
>
> ## Question 3: Regarding ReAG construction: Have you performed any analysis on the accuracy, etc.
>
> For systems that support well-structured trajectories, parsing issues are basically nonexistent. We validate the parser's ability to analyze non-structured trajectories.
> We categorize the parsed differences between the parsed steps and the original ground truth steps into three types:
>
> 1. Missing sentence(s)
>
> 2. Step split – one original step is split into two or more steps without losing any content
>
> 3. Step merge – two or more original steps are merged, sometimes with content missing
>
> To evaluate our parser, we randomly sampled 30 cases from our dataset and concatenated their originally step-wise separated conversation histories into a single continuous sequence. We then applied our trace parser to re-segment this sequence into discrete steps and conducted a manual evaluation to assess its performance.
>
> The total number of original steps was 657. The differences observed were:
>
> - Missing content: 29 steps (4.41%)
>
> - Split steps: 9 steps (1.37%)
>
> - Merged steps with missing content: 11 steps (1.67%)
>
> - Merged steps without missing content: 16 steps (2.44%)
>
> Overall, 9.89% of the steps differed from the original, and 6.09% contains missing content. Furthermore, since most agent frameworks support structured trajectory logging, this proportion would be significantly lower in practical applications.

---

> ### Author Response · Authors · 2025-12-04
> **Updated PDF for Reviewer Feedback**
>
> Dear reviewer,
> We have uploaded the revised PDF addressing all reviewer comments. We kindly invite you to take a look at the updated version. Thank you for your time and constructive feedback.

---

### Official Review · Reviewer_G1Tb · 2025-10-31

**Soundness:** 3
**Presentation:** 3
**Contribution:** 3
**Rating:** 6
**Confidence:** 4

**Summary:**

This paper proposes AGENTSCOPE, a neuro-symbolic framework for diagnosing failure modes in LLM-based agents. The key idea is to abstract agent trajectories into a Reasoning-Action Graph (ReAG), then check failures via a set of neural invariants that formalize different error types (e.g., incorrect context, instruction un-following, faulty reasoning, planning failures). AGENTSCOPE performs both failure localization and failure attribution, while existing works mainly treat LLM as judge and often cannot pinpoint error step or category reliably.
The authors evaluate on Who&When and a newly created benchmark AgentErrata (generated via failure-taxonomy-guided injection). Results show AGENTSCOPE significantly outperforms previous LLM-as-judge or step-by-step heuristics in both localization and classification accuracy.

**Strengths:**

1. The paper addresses agent failure diagnosis, which is extremely important right now. With everyone building agent frameworks (LangChain, AutoGen, OpenManus, etc.), debugging becomes the biggest bottleneck. So the problem choice is high-impact. The idea of combining symbolic invariants + neural semantic checks is novel. Most recent works simply treat GPT-4 as judge or fine-tune a classifier on trajectories.
2. I like the taxonomy in Table 1. This formulation is also a good contribution. The breakdown is meaningful: real agent failures often mix context issues + planning derail. This taxonomy captures that.
3. The authors evaluate on two real datasets (Who&When, AgentErrata)  and test on multiple foundation models: GPT-4o, DeepSeek-V3, Qwen-14B. The experiments demonstrate the performance.

**Weaknesses:**

1. Some ablations are needed. You may need to show the effect of different components in your method.
2. How does AGENTSCOPE handle adversarial agents, e.g., self-masking behavior (LLM intentionally hides chain-of-thought)?
Paper mentions reward hacking, but no experiments. For example, If the executed tool call returns wrong data on purpose, can system detect?

**Questions:**

How does the method handle adversarial agents?

---

> ### Author Response · Authors · 2025-11-21
>
> Thank you for the detailed and constructive feedback! We treasure the opportunity to address your concerns and improve our work.
>
> ## Weakness 1: Some ablations are needed, etc.
>
> Thank you for pointing out this weakness.
>
> 1. **Effectiveness of the parser**
>
>     For systems that support well-structured trajectories, parsing issues are basically nonexistent. We validate the parser's ability to analyze non-structured trajectories.
>
>     We categorize the parsed differences between the parsed steps and the original ground truth steps into three types:
>
>    1) Missing sentence(s)
>
>    2) Step split – one original step is split into two or more steps without losing any content
>
>    3) Step merge – two or more original steps are merged, sometimes with content missing
>
>     To evaluate our parser, we randomly sampled 30 cases from our dataset and concatenated their originally step-wise separated conversation histories into single continuous sequences. We then applied our trace parser to re-segment these sequences into discrete steps and conducted a manual evaluation to assess the trace parser’s performance.
>
>     The total number of original steps was 657. The differences observed were:
>
>    - Missing content: 29 steps (4.41%)
>
>    - Split steps: 9 steps (1.37%)
>
>    - Merged steps with missing content: 11 steps (1.67%)
>
>    - Merged steps without missing content: 16 steps (2.44%)
>
>    Overall, 9.89% of the steps differed from the original, and 6.09% contains missing content. Furthermore, since most agent frameworks support structured trajectory logging, this proportion would be significantly lower in practical applications.
>
> 2. **Effectiveness of the two stages diagnosis process**
>
>    The effectiveness of the two stages is demonstrated in Table 3 and Table 4 of the paper. In the CA (classification accuracy) column, the results before the slash (/) indicate  “w/o IV”, that is to say they were performed by the preliminary prediction stage only; and the results after the slash (/) indicate “w/ IV”, which were obtained by the preliminary prediction stage followed by the secondary verification stage. The results show that the preliminary prediction stage itself can achieve acceptable prediction accuracy, and that the secondary verification stage, though having little impact on the agent level and step level accuracy, can significantly boost the classification accuracy of failure mode.
>
> ## Weakness 2: How does AGENTSCOPE handle adversarial agents, e.g., self-masking behavior (LLM intentionally hides chain-of-thought), etc.
>
> This "weakness" issue will be combined with question 1 for a single reply.
>
> ## Question 1: How does the method handle adversarial agents?
>
> Thank you for raising this point. AgentScope currently follows the standard assumption adopted in existing agent-evaluation research [Zhang et al., 2025b, Cemrietal.,2025, etc.] the agent being analyzed is non-adversarial. Our method therefore does not explicitly address adversarial behaviors at this stage.
>
> We agree that robustness against adversarial agents is an important direction. Extending AgentScope to handle adversarial or intentionally deceptive agents is a valuable line of future work, and we will discuss this limitation and its potential extensions in the revised version.

---

> > ### Comment · Reviewer_G1Tb · 2025-11-26
> > **Response**
> >
> > Thanks to the authors for the rebuttal. I will keep my score. My concerns are addressed.

---

> ### Author Response · Authors · 2025-12-04
> **Updated PDF for Reviewer Feedback**
>
> Dear reviewer,
> We have uploaded the revised PDF addressing all reviewer comments. We kindly invite you to take a look at the updated version. Thank you for your time and constructive feedback.

---

### Official Review · Reviewer_AgWR · 2025-10-31

**Soundness:** 2
**Presentation:** 1
**Contribution:** 2
**Rating:** 4
**Confidence:** 3

**Summary:**

This work proposes a neuro-symbolic approach for agent failure attribution, named AgentScope. The key idea is to leverage LLM-guided reasoning atop the structured representation against the so-called neural invariants. An extra benchmark, AgentErrata, is further introduced which contains failure type annotations and is obtained by fault injection. Experiments on Who&When and AgentErrata show generally improved performance with the proposed method.

**Strengths:**

1. Good empirical performance

**Weaknesses:**

My major concern is the clarity of technical details of the proposed method. Specifically, I started to get lost in Section 3.2. While "neural invariants" is featured as a novel concept and key point in AgentScope, the definition (either formal or informal) is not given. Furthermore, how exactly the LLM is applied upon neural invariants to localize and classify failures is not discussed either. Lastly, as another example, in line 239-240, "the second stage (secondary verification by Invariant Verifier, or IV) aims to reduce false positives." What is this "Invariant Verifier"? In all, currently I'm afraid the key ideas and components of the proposed method remain unclear.

**Questions:**

1. In Table 2, is the base LLM the same for AgentScope and baselines?

---

> ### Author Response · Authors · 2025-11-21
>
> Thank you for the detailed and constructive feedback! We treasure the opportunity to address your concerns and improve our work.
>
> ## Weakness 1: My major concern is the clarity of technical details of the proposed method, etc.
>
> Thank you for the feedback! We will add the definitions and procedural details of several key components—such as neural invariants, how LLMs are applied to evaluate them, and the role of the Invariant Verifier (IV) in the new version.
> Specifically, we will add:
>
> - clear formal and informal definitions of neural invariants,
> - a step-by-step description of how neural invariants are evaluated by LLMs for failure localization and classification, and
> - an explicit explanation of the Invariant Verifier and its role in reducing false positives.
>
> These details will be included in the appendix and referenced from the main text for clarity. We are preparing an updated version to be released on Dec 3, and please check it out once available.
>
> ## Question 1: In Table 2, is the base LLM the same for AgentScope and baselines, etc.
>
> Yes, the base LLM used for AgentScope and all baselines in Table 2 is the same. We will clarify this explicitly in the revised version.

---

> ### Author Response · Authors · 2025-12-04
> **Updated PDF for Reviewer Feedback**
>
> Dear reviewer,
> We have uploaded the revised PDF addressing all reviewer comments. We kindly invite you to take a look at the updated version. Thank you for your time and constructive feedback.

---

### Official Review · Reviewer_DoKc · 2025-11-04

**Soundness:** 3
**Presentation:** 3
**Contribution:** 3
**Rating:** 4
**Confidence:** 3

**Summary:**

This paper presents AGENTSCOPE, a neuro-symbolic framework designed to diagnose failures in LLM agents. The authors argue that existing methods, such as traditional software debugging or holistic "LLM-as-a-judge" approaches, are insufficient for pinpointing failures in complex agent trajectories. The core methodology of AGENTSCOPE involves two main steps: (1) Behavioral Abstraction and (2) Invariant Verification. To evaluate this framework, the authors created a new benchmark dataset, AgentErrata, using failure-taxonomy-guided fault injection. Experiments on AgentErrata and the existing Who&When dataset show that AGENTSCOPE outperforms baselines in both failure localization and failure attribution.

**Strengths:**

1. The paper tackles a critical and challenging problem. As LLM agents become more autonomous and their interaction trajectories grow longer, the need for scalable, reliable, and interpretable debugging tools is paramount

2. The idea of abstracting complex, unstructured agent trajectories into a formal Reasoning-Action Graph (ReAG) is a strong and well-motivated contribution. This structured representation is more amenable to formal analysis than raw logs.

3. The authors create and contribute the AgentErrata dataset, a new benchmark valuable to the community. This dataset is necessary for developing and evaluating diagnostic tools because, unlike previous datasets, it provides annotations for both the failure steps and their explicit failure types (categories) .

**Weaknesses:**

1.  While AGENTSCOPE shows relative improvement over the baselines, its absolute performance is very low, casting doubt on its practical utility. On the new AgentErrata dataset, the framework achieves only 30.48% Step-Level Accuracy (SLA) and 19.05% Classification Accuracy (CA). This means the tool fails to identify the exact failure step 70% of the time and fails to correctly classify the failure type 81% of the time. These low numbers suggest the framework is not yet effective or reliable enough for real-world diagnosis.

2.  The baselines used for comparison seem weak. For attribution (Table 3), the "LLM-as-Judge" baseline achieves only 5.35% SLA and 4.40% CA. This is surprisingly low for a model like GPT-4o, suggesting a potentially poor implementation or prompt. A stronger baseline would be to provide the full ReAG and the failure taxonomy (Table 1) to the LLM judge, which is not tested.

3. Table 2 presents results for "Method w/ GT" and "Method w/o GT" without any explanation in the text or captions as to what with ground truth or without groudtruth means.

4. The Concept of "Neural Invariant" is vague, the paper claims NIs are specified with "neural functions". However, Appendix A reveals these are symbolic logic predicates that rely on sub-functions like sim(), relevant(), consistent(), etc. The paper states that checking these functions often relies on an LLM-as-judge. This seems to contradict the paper's initial premise, which heavily criticizes LLM-as-judge approaches for being unreliable.

**Questions:**

1. Could you please provide a precise definition of a "neural function" (e.g., is_valid(), consistent(), relevant())? Are these fine-tuned models, or are they, as the appendix suggests, simply calls to a general-purpose LLM (like GPT-4o) with a specific prompt?


2. If these "neural functions" are indeed LLM calls, how do you reconcile this with your core criticism that "relying on LLMs as the judge yields unreliable diagnosis results"? How does AGENTSCOPE fundamentally overcome the unreliability of LLM judges, rather than just applying them at a different granularity?


3. Please clarify the distinction between "Method w/ GT" and "Method w/o GT" in Table 2. What "Ground Truth" information is being used, and why does this lead to such different results?

---

> ### Author Response · Authors · 2025-11-21
>
> Thank you for the detailed and constructive feedback! We treasure the opportunity to address your concerns and improve our work.
>
> ## Weakness 1: While AGENTSCOPE shows relative improvement over the baselines, etc.
>
> Thank you for the insightful comment. We acknowledge that the current results on AgentErrata show room for improvement. We would like to clarify that some of the reported numbers were affected by issues in the earlier version of the dataset and evaluation pipeline. We are correcting these issues (For example, the failure is not aligned with the context.) and are generating updated results.
> We will provide partial updated results first and a fully revised version of the paper with complete corrected numbers by Dec 3. Our preliminary findings indicate that the performance improves notably after fixing the issues. We appreciate the reviewer’s feedback and will ensure that the final revision presents more accurate and up-to-date results.
> Currently, we sampled 52 traces from AgentErrata and made a preliminary refinement, here are results:
>
> |data version|ALA | SLA | CA |
> |:-:|:-:|:-:|:-:|
> | original | 47.17% | 33.96% | 13.21% |
> | refined | 51.92% | 48.08% | 21.15% |
>
> (the experiment settings were: GPT-4o as base LLM model, without ground truth of the original problem, with Invariant Verifier)
>
> Due to time constraints, we are currently only testing on a small set. We believe that the full experiment will be run in the next revised version, and issues will be fixed with further improvements.
>
> ## Weakness 2: The baselines used for comparison seem weak, etc.
>
> The “LLM-as-Judge” baseline in Table 3 uses the original prompt in the Who&When paper [Zhang et al., 2025b].
> Therefore, unexpectedly poor results reflect the benchmark’s default configuration, not an implementation issue on our side.
> We agree that exploring stronger LLM-judge baselines—such as supplying the full ReAG context—would be valuable. We plan to run these enhanced baselines and will include the updated results in the revised version.
>
> ## Weakness 3: Table 2 presents results for "Method w/ GT" and "Method w/o GT" without any explanation, etc.
>
> This "weakness" issue will be combined with question 3 for a single reply.
>
> ## Weakness 4: The Concept of "Neural Invariant" is vague, etc.
>
> This "weakness" issue will be combined with question 1 for a single reply.

---

> ### Author Response · Authors · 2025-11-21
>
> ## Question 1: Could you please provide a precise definition of a "neural function", etc.
>
> The “neural functions” (e.g., is_valid(), consistent(), relevant()) are not fine-tuned models. As indicated in the appendix, currently, each neural function is implemented as a call to a general-purpose LLM (e.g., GPT-4o) with a task-specific prompt. No additional training or fine-tuning is performed. Note that the implementation of neural functions can be customized, e.g., using fine-tuned models.
> We will add a clear and explicit definition of neural functions in the revised version to avoid ambiguity.
>
> For example, the following shows the prompt of a neural function implementation (simNode()) for the failure mode “Step Repetition”.It is necessary to traverse adjacent nodes in the graph and apply the current function to each pair of nodes.
>
> prompt = (
>
>     "You are given two nodes from a reasoning execution graph representing an AI agent's problem-solving process. Judge whether the two nodes are semantically similar or not. Return only 'Yes' (for similar) or 'No' (for not similar). The nodes are considered similar if they convey the same meaning, perform the same function, or represent the same step in the reasoning process, even if they are worded differently. Conversely, if the nodes have different meanings, functions, or represent different steps, they are considered not similar. Even subtle similarities should lead to a 'Yes' answer.\n"
>     f"Node 1: {node1}\n"
>     f"Node 2: {node2}\n"
>     "Your response should be either 'Yes' or 'No' without any additional explanation."
>
> )
>
> ## Question 2: If these "neural functions" are indeed LLM calls, etc.
>
> Thank you for the question. Our key point is not to criticize using LLMs per se, but the all-at-once evaluation paradigm in which the LLM is asked to directly judge the entire trajectory or classify the failure in a single step. Prior work shows that this holistic judging is unstable and sensitive to prompt variance.
> AGENTSCOPE addresses this by introducing a structured execution process and a factorized error taxonomy. Instead of asking the LLM to produce a full diagnosis in one opaque step, we decompose the reasoning process into interpretable substeps and apply small, localized neural functions (e.g., is_valid, consistent, relevant) that evaluate specific, well-scoped properties. Neural behaviors still require neural methods to assess, but by translating neural evaluations into symbolic, step-level signals, the final diagnosis becomes transparent, compositional, and verifiable.
> In short, AGENTSCOPE does not rely on a monolithic LLM-as-judge. It leverages LLMs only for specific, well defined subdecisions, and integrates them within a structured symbolic reasoning pipeline—substantially improving reliability.
>
> ## Question 3: Please clarify the distinction between "Method w/ GT" and "Method w/o GT" in Table 2, etc.
>
> Sorry for not spelling them out. The “w/ GT” and “w/o GT” in Table 2 follow the definitions from the Who&When paper [Zhang et al., 2025b]. Specifically, w/ GT corresponds to the Impact of Ground Truth on Failure Attribution, where the evaluator is provided with the ground-truth answer to the original problem that the agent system in the conversation history was solving, while w/o GT refers to the setting where no ground-truth answer is available. We will add the definition in the paper.

---

> ### Author Response · Authors · 2025-12-04
> **Updated PDF for Reviewer Feedback**
>
> Dear reviewer,
> We have uploaded the revised PDF addressing all reviewer comments. We kindly invite you to take a look at the updated version. Thank you for your time and constructive feedback.

---

### Meta-Review · Area_Chair_cDk5 · 2026-01-03

**Summary:**

This paper presents AGENTSCOPE, a neuro-symbolic framework designed to diagnose failures in LLM agents. Reviewers provide 4,4,6,6. Reviewers shares some concerns such as details, low absolute accuracy, LLM-as-judger evaluation and so on. (The details can be found in reviewer concerns section. ). AC thinks most concerns have been addressed. However, AC shared the similar concern regarding to the low absolute performance. Authors have conducted new experiments on a small set and said will provide a result on Dec 3. However, AC did not see any updated results on it. AC feels this concern is valid and does not think the reviewer DoKc will increase the score based on it. Thus, AC hopes the authors can improve the paper and add new results in the revised version.

**Reviewer Concerns:**

Reviewer DoKc

Main concerns

- Low absolute performance (esp. CA ~19% originally) questions practical utility.

Authors have clarified that some of the results were affected by issues in the earlier version of the dataset and evaluation pipeline. They conducted new experiments on a small sample of traces from agentErrata. Authors said a fully revised version of the paper with complete corrected numbers by Dec 3. But AC did not find the results.

- Baselines may be weak.

Authors used the same version from the [zhang 2025b] and agree that exploring stronger LLM-judge baselines would be valuable.

Unclear “Method w/ GT vs w/o GT” in Table 2.

Authors have addressed the concerns by providing explanations.

AC thinks most of the concerns have been addressed. The main concerns left are low absolute performance. The improved metrics were shown only on 52 traces (partial); reviewers may still question whether the updated improvements hold across the full dataset and all settings.

Reviewer AgWR

Main concerns: missing/unclear details.

Authors have provided the details. AC thought the concern had been fully addressed.


Reviewer G1Tb

Main concerns

1) Requested ablations

Authors have provided the ablations.

2) Adversarial agents.

Authors follow the standard existing agent-evaluation research on non-adversarial settings. Authors did not explain the performance against adversarail agents. AC feels it should at least mention it and add a limitation section to discuss it, since authors have mentioned reward hacking. But overall, AC does not think the reviewer will decrease the score based on it.

Reviewer gdmo

Main concerns

1) missing details

Authors have added the details. It sounds reasonable.

2). Low absolute CA.

Authors conducted experiment on a small set.

3). ReAG construction
Authors have explained it well.


Overall, most concerns have been addressed except for the low absolute performance.

**Reviewer Scores:**

Reviewer DoKC will not change the score since the concern regarding to the low absolute performance is not fully addressed.
Reviewer AgWr will increase the score since the concerns have been fully addressed.
Reviewer G1Tb  should not descrease the score since adversarial agent setting is acceptable limitation under the scope of this paper. But Ac still suggests the authors adding the discussion.
Reviewer gdmo shared the similar concerns regarding to the performance. AC feels the rebuttal does not fully addressed it.

---

### Decision · Program_Chairs · 2026-01-26

Reject